# Prediction with expert advice under additive noise

**Alankrita Bhatt**
Granica Computing, Inc.
alankrita.112@gmail.com

**Victoria Kostina**
California Institute of Technology
vkostina@caltech.edu

## Abstract

Prediction with expert advice serves as a fundamental model in online learning and sequential decision-making. However, in many real-world settings, this classical model proves insufficient as the feedback available to the decision-maker is often subject to noise, errors, or communication constraints. This paper provides fundamental limits on performance, quantified by the regret, in the case when the feedback is corrupted by an additive noise. Our general analysis achieves sharp regret bounds for canonical examples of such additive noise as the Gaussian distribution, the uniform distribution, and a general noise with a log-concave density. This analysis demonstrates how different noise characteristics affect regret bounds and identifies how the regret fundamentally scales as a function of the properties of the noise distribution.

## 1 Introduction

The prediction with expert advice framework is a cornerstone of online learning and sequential decision-making [CBFH$^+$97, CBL06, H$^+$16, Ora19]. In this setting, a decision-maker repeatedly selects an action over a sequence of rounds, leveraging the recommendations of a finite collection of "experts." At each round, the decision-maker may choose one expert's action or form a mixture over them. After observing the losses incurred by all experts, the learner updates its decision rule to guide future actions. The overarching goal is to ensure that the learner's cumulative loss remains close to that of the best single expert in hindsight. Because this framework abstracts a broad range of applications in domains such as finance, online advertising, and game playing, it continues to serve as one of the most influential paradigms in online learning.

Despite its simplicity and generality, the classical expert setting assumes the decision-maker observes exact feedback in the form of the experts' losses. However, this assumption is often unrealistic in practical environments, where feedback can be noisy, incomplete, or rate-limited. Consider, for instance, autonomous driving: decision-making is constrained by the time and bandwidth needed to process sensor data, while the sensory inputs themselves may be corrupted by noise from environmental or hardware factors. Similar challenges arise in financial markets, where imperfect information distorts feedback signals. In such cases, the learner must adapt to uncertainty not only from the environment, represented by the adversarially chosen losses, but also from the noise affecting feedback. This motivates the study of algorithms that can learn effectively under imperfect observations, maintaining robustness while still achieving low regret.

We now formalize the standard prediction with experts framework before extending it to noisy feedback. Let there be $m$ experts and a time horizon of $n$ rounds. At each round $t \in [n]$:

- The decision-maker selects a probability distribution $p_t \in \Delta^{m-1}$ over the experts, based on all past observations. This can be viewed as assigning a weight to each expert.
- The adversary then reveals a loss vector $\ell_t \in \mathcal{L} := [0,1]^m$, where $\ell_{tj}$ denotes the loss of expert $j$ at round $t$. The learner's expected loss for that round is $\langle p_t(\ell^{t-1}), \ell_t \rangle$, i.e., the average loss under the chosen mixture (where $\langle \cdot, \cdot \rangle$ denotes the standard inner product).

39th Conference on Neural Information Processing Systems (NeurIPS 2025).

A strategy $p$ thus corresponds to a sequence of mappings $p_t(\cdot)_{t=1}^n$, with $p_t : \mathcal{L}^{t-1} \to \Delta^{m-1}$. Given any sequence of outcomes $\ell^n = (\ell_1, \dots, \ell_n)$, the learner's regret is defined as

$$\mathrm{Reg}(p, \ell^n) := \sum t = 1^n \langle p_t(\ell^{t-1}), \ell_t \rangle - \min_{j \in [m]} \sum_{t=1}^n \ell_{tj}. \tag{1}$$

In words, the regret quantifies how much worse the learner performs compared to the best fixed expert chosen in hindsight—an expert whose identity cannot be known until the sequence ends. The learner's challenge is thus to minimize this gap causally, adapting its choices as it recieves more feedback.

A central quantity of interest is the minimax regret, defined as the smallest possible regret achievable by any strategy against the most adversarial sequence of losses:

$$\inf_p \sup_{\ell^n} \mathrm{Reg}(p, \ell^n).$$

A classical result [CBFH$^+$97] establishes a sharp characterization of this value:

$$\inf_p \sup_{\ell^n} \mathrm{Reg}(p, \ell^n) = \Theta(\sqrt{n \log m}), \tag{2}$$

where the notation $\Theta(\cdot)$ hides universal constants independent of $n$ and $m$. This $\sqrt{n \log m}$ scaling represents the optimal rate for learning with expert advice, setting the benchmark for subsequent extensions to more complex feedback models.

We now describe the *prediction with noisy expert advice* setting, in which the decision-maker does not observe the true expert losses directly, but instead receives a *corrupted* or *partial* observation $c_t$ of the loss vector $\ell_t$. For instance, $c_t$ might represent a noise-perturbed version of $\ell_t$, or a quantized signal produced when the loss is transmitted through a rate-limited communication channel. Formally, the noisy feedback model consists of two components:

- **Channel:** a (possibly stochastic) mapping applied to the sequence of losses, $c_t : \mathcal{L}^t \to \mathcal{C}$, where $\mathcal{C}$ denotes the output alphabet of the channel. This transformation may depend on the current and past losses and introduces the noise or compression governing the feedback.

- **Decision rule:** at each round, the learner selects a probability distribution $p_t(c^{t-1})$ over experts, based solely on the previously observed channel outputs $c^{t-1}$.

The central difficulty in this framework lies in the fact that the learner must compete against the best expert with respect to the *true* (uncorrupted) losses $\ell_t$, despite only observing the degraded signals $c_t$. Accordingly, we define the regret in this setting as

$$\mathrm{Reg}(p, \mathsf{P}_{c|\ell}, \ell^n) \triangleq \sum_{t=1}^n \langle \mathsf{E}[p_t(c^{t-1})], \ell_t \rangle - \min_{j \in [m]} \sum_{t=1}^n \ell_{tj}, \tag{3}$$

where the expectation is taken with respect to the randomness of the channel $P_{c|\ell}$.

This formulation generalizes the standard prediction-with-experts setup (cf. (1)). While the benchmark term (the cumulative loss of the best fixed expert $\min_{j \in [m]} \sum_{t=1}^n \ell_{tj}$) remains identical, the learner now faces the additional challenge of operating under information degradation. As a result, the achievable regret depends not only on the underlying loss sequence and the learner's strategy but also on a measure of the channel's fidelity.

While the outlined noisy expert advice setting is quite general, the practically-motivated class of *additive-noise channels* is of particular interest in this paper. This is a subset of the class of channels with memoryless noise, where the $c_t$ is the output of a *fixed known* random transformation $\mathsf{P}_{c|\ell}$ with input $\ell_t$. In particular, the output $c_t = \ell_t + Z_t$ where $Z_t$ is drawn from some fixed distribution. In this case, we wish to devise a decision strategy $p$ that at time step $t$ maps the noisy outputs $c^{t-1}$ to a decision and achieves low regret as defined in (3).

The study of additive noise channels is motivated by numerous real-world applications where feedback is corrupted by predictable noise patterns. Gaussian additive noise represents perhaps the most ubiquitous noise model, arising naturally in settings such as sensor networks, where thermal noise in electronic components follows a Gaussian distribution; financial market predictions, where price

observations contain normally distributed measurement errors; and healthcare monitoring systems, where physiological measurements are subject to Gaussian instrument noise. Uniform additive noise, on the other hand, is particularly relevant in scenarios involving quantization and digital conversion, such as in communication-constrained IoT networks where continuous signals must be discretized with limited precision; in crowdsourced data collection where human bias introduces bounded random errors; and in privacy-preserving systems where uniform noise is deliberately added to protect sensitive information while maintaining utility. Both noise models represent fundamental corruption patterns that algorithms must contend with when learning from imperfect feedback, making their theoretical analysis particularly valuable for designing robust decision-making systems.

## 2  Main results

Our primary contribution is a comprehensive characterization of the fundamental limits of prediction with expert advice under various additive noise channel models. These results also provide practical algorithmic insights for real-world applications where feedback is inherently noisy. In particular, our results specialize to the following canonical additive noise models to yield the following results:

- **Gaussian noise:** For Gaussian noise with variance $\sigma^2$ (denoted by AWGN($\sigma$))

$$\mathrm{Reg}(\mathrm{AWGN}(\sigma)) = \Theta\left(\sqrt{(1+\sigma^2)n \log m}\right) \tag{4}$$

  This quantifies how increasing noise variance directly impacts the achievable regret, with a clean dependency on the standard deviation.

- **Uniform noise:** When feedback is corrupted by uniform noise distributed in $[-\sigma, \sigma]$ (denoted by AddUnif($\sigma$))

$$\mathrm{Reg}(\mathrm{AddUnif}(\sigma)) = \Theta\left(\sqrt{(1+\sigma)n \log m}\right) \tag{5}$$

  This result is particularly interesting as it reveals a fundamentally different scaling with the noise parameter compared to the Gaussian case. The linear rather than quadratic dependence on $\sigma$ highlights how the shape of the noise distribution—not just its variance—critically affects learning performance.

- **Symmetric log-concave noise:** Extending beyond specific distributions, we provide bounds for the general class of symmetric log-concave noise distributions with variance $\sigma^2$ (denoted by Add($f_\sigma$))

$$\Omega\left(\sqrt{(1+\sigma)n \log m}\right) \le \mathrm{Reg}(\mathrm{Add}(f_\sigma)) \le O\left(\sqrt{(1+\sigma^2)n \log m}\right) \tag{6}$$

  Symmetric log-concave distributions are particularly valuable to study as they encompass many real-world noise models in sensor networks, signal processing, and privacy-preserving systems. In particular, they encompass the previous two examples of Gaussian and Uniform distribution, and also include other noise distributions such as the Laplace (double exponential) distribution. This broad class maintains favorable concentration properties while often better modeling the heavier tails observed in practical measurement errors compared to purely Gaussian assumptions, making our bounds widely applicable to realistic noise scenarios. While the characterization (6) is not perfectly tight, it encompasses both previous results as special cases: the lower bound is tight for uniform noise, and the upper bound is tight for Gaussian noise.

Our results reveal that as noise intensity increases ($\sigma \to \infty$), the regret eventually grows superlinearly with time horizon $n$, indicating the fundamental impossibility of learning effectively when feedback becomes extremely corrupted. Conversely, our bounds smoothly transition to the classical noiseless case as noise diminishes. These theoretical guarantees are derived through an analysis combining two complementary techniques: (1) an enhanced exponential weights algorithm adapted for noisy feedback, and (2) information-theoretic lower bounds that precisely characterize what is fundamentally impossible to achieve. In particular, we employ the following two results:

**Theorem 1 (informal)** *For any memoryless feedback channel*

$$Regret \le O\left(\frac{\log m}{\alpha} + \alpha \cdot n \cdot MSE_{estimation}\right)$$

*where $m$ is the number of experts, $n$ is the time horizon, $\alpha$ is a learning rate parameter, and $MSE_{estimation}$ represents the mean squared error in estimating the true losses from noisy observations.*

We note, in particular, that the regret additionally depends on the mean squared error (MSE) obtained by the estimator $\widehat{\ell}_t$, drawing an interesting connection between estimation and noisy regret minimization. This result echoes the well-known separation principle in measurement feedback control [KSH00], where optimal control cost can be decomposed into optimal estimation cost followed by optimal control cost based on the estimates. Our theorem suggests a similar phenomenon in online learning: the performance degradation due to noisy feedback can be directly quantified through estimation error.

**Theorem 2 (informal)** *In memoryless channels*

$$Regret \geq \Omega \left( \sqrt{\frac{n \cdot \log m}{\eta(\mathsf{P}_{c|\ell})}} \right)$$

*where $\eta(\mathsf{P}_{c|\ell})$ is the strong data-processing constant of the channel—an information-theoretic measure quantifying how well the channel preserves information.*

This lower bound demonstrates that the fundamental barrier to learning is information-theoretic in nature: as the channel's ability to transmit information degrades (smaller $\eta$), the minimum achievable regret increases proportionally.

## 3   Related work

In this section, we summarize prior work that relates most closely to the problem studied in this paper. For brevity, we focus only on the works that are most directly connected to our setting, and defer a more exhaustive literature overview to Appendix A.

To our knowledge, the earliest study of prediction with noisy feedback for individual loss sequences $\ell^n$ was by Weissman, Merhav, and Somekh-Baruch [WMSB01], following the foundational works of [FMG92, MF98]. Their analysis focused on the case where the losses are corrupted by a binary symmetric channel (BSC). They introduced upper bounds and a notion of conditional finite-state predictability, and proposed universal prediction schemes that are robust both to the choice of the best expert in hindsight and to the unknown channel bias. Subsequent work by Weissman and Merhav [WM01] generalized these ideas to a broader class of universal prediction schemes for noisy individual sequences, while [WM04] extended the analysis to noisy prediction of stationary ergodic sources. However, these results do not directly cover additive-noise channels, nor do they establish matching lower bounds on regret. The study in [WM01] was later extended by Resler and Mansour [RM19] to the adversarial bandit framework—where only the loss of the chosen expert is revealed at each round—though their setting also assumed a BSC and is therefore not applicable to additive-noise models.

In recent years, motivated partly by the rise of federated learning as a key paradigm in distributed optimization [KMA+21], there has been growing interest in decision-making under communication or rate constraints. Similar questions have been explored in the context of stochastic bandits [HYF22, MHP23, MST23], which also served as inspiration for our work. In these studies, the focus is on quantifying how limited feedback precision affects achievable regret. Specifically, for multi-armed and linear bandit problems, [HYF22] and [MHP23] proposed communication-efficient algorithms that attain regret comparable to the full-precision case while characterizing the number of bits required to do so. Mayekar et al. [MST23] considered an additional constraint, modeling the feedback channel as a power-limited AWGN link, and derived both achievability and converse bounds showing that the regret deteriorates by a multiplicative factor of $\sqrt{1/\mathsf{SNR}}$, where SNR denotes the signal-to-noise ratio. Their analysis, however, is limited to Gaussian channels and relies on a UCB-style algorithm that is not applicable in our framework.

In contrast, this paper addresses the *full-information* (experts) setting with *adversarial* losses for individual sequences. We establish near-optimal upper and lower bounds on the regret and specialize our results to a general family of additive-noise channels. Our findings unify and extend several existing results: for instance, in the cases of one-bit per-expert quantization and AWGN noise, our

regret bounds coincide (up to constants) with those in [HYF22, MST23]. Finally, [BK24] recently investigated prediction with noisy expert advice, but their analysis has notable limitations that our work overcomes. Specifically, we handle both bounded and unbounded loss functions—an essential feature for dealing with additive noise that may have heavy tails. Moreover, whereas their Gaussian lower bound was stated without proof, our result derives it rigorously from a unified framework (Theorem 2). Most importantly, our analysis provides tight or near-tight characterizations for a broad range of additive-noise channels, including uniform and symmetric log-concave distributions, which are not captured by their framework.

## 4 Upper bounds

In this Section, we provide upper bounds for prediction with noisy expert advice with additive noise. To do this, we need to construct a decision-making strategy and prove its performance limits. To this end, consider the following estimator:

$$p_{tj}^{\mathrm{EW}}(c^{t-1}) \propto \exp\left(-\alpha \sum_{i=1}^{t-1} \widehat{\ell}_{ij}\right). \tag{7}$$

where $\widehat{\ell}_t$ is any function $f(c_t)$ satisfying $\mathsf{E}[f(c_t)] = \ell_t$. In other words, the employed strategy (which we denote by $\widehat{p}^{\mathrm{EW}}$) is simply the landmark exponential weights strategy [CBFH+97] that is known to be optimal in a sense for the vanilla prediction with expert advice problem [CBL06] used in conjunction with an unbiased estimator for the true loss $\ell_t$, with a fixed learning rate $\alpha$. Unbiasedness of an estimator is an important property in statistical estimation, with several interesting and attractive consequences [LC06, Chapter 2]. In the realm of online learning, one of the prominent examples of the use of unbiased estimator as a proxy for an unknown loss is in the celebrated EXP3 algorithm of [ACBFS95].

We can now state Theorem 1 formally.

**Theorem 1** *Let the channel $P_{c|\ell}$ be memoryless and let $\widehat{\ell}_t$, constructed using $c_t$, be an unbiased estimator. For any $\alpha > 0$ defining the event*

$$\mathcal{E} := \{\exists t, j : -\alpha\widehat{\ell}_{tj} \geq 1\}, \tag{8}$$

$$\mathrm{Reg}(\widehat{p}^{\mathrm{EW}}, P_{c|\ell}, \ell^n) \leq \frac{\log m}{\alpha} + \alpha n \left(1 + \max_{j,t} \mathsf{E}[\ell_{tj} - \widehat{\ell}_{tj}]^2\right)$$
$$+ \sqrt{4m^2 n^2 \left(1 + \max_{t,j} \mathsf{E}[\widehat{\ell}_{tj} - \ell_{tj}]^2\right) \mathbb{P}(\mathcal{E})}.$$

We once again point out that the regret depends on the mean squared error (MSE) obtained by the estimator $\widehat{\ell}_t$, drawing an interesting connection between estimation and noisy regret minimization. Theorem 1 follows from a "second-order" analysis of the exponential weights strategy $p^{\mathrm{EW}}$ which bounds the regret incurred by $p^{\mathrm{EW}}$ in terms of the second moment of the loss functions [CBMS07, GSVE14]. It follows the standard idea of constructing a potential function and carefully bounding the change in the potential function at each time step, and the full proof is relegated to Appendix B.

### 4.1 Application of Theorem 1 to canonical channel models

We apply our general upper bound from Theorem 1 to several important additive noise channels. For each channel model, we develop an appropriate unbiased estimator, calculate its estimation error variance, and determine the resulting regret guarantees by substituting these values into the framework established in Theorem 1. Recall that for additive noise channels, the output $c_{tj} = \ell_{tj} + Z_{tj}$ where all the $Z_{tj}$ are independently and identically distributed.

**Gaussian noise.** Consider $c_t = \ell_t + Z_t$ where $Z_t \sim \mathcal{N}(0, \sigma^2 I)$. The most natural unbiased estimator to use is simply $\widehat{\ell}_t = c_t$, with MSE $\mathsf{E}[(c_t - \ell_t)^2] = \sigma^2$. Note that in this case since the noise is unbounded $-\alpha c_{tj}$ can be arbitrarily large—but the probability of this event occuring is exponentially

small. In particular, recalling from (8) that the event $\mathcal{E}$ is defined as $\mathcal{E} := \{\exists t, j : -\alpha c_{tj} \geq 1\}$ we note for $\alpha = \sqrt{\frac{\log m}{n(1+\sigma^2)}}$

$$\mathbb{P}(\mathcal{E}) \overset{(a)}{\leq} \sum_{t=1}^{n} \sum_{j=1}^{m} \mathbb{P}\left(c_{tj} \leq -\sqrt{\frac{n(1+\sigma^2)}{\log m}}\right)$$

$$\overset{(b)}{\leq} \sum_{t=1}^{n} \sum_{j=1}^{m} \mathbb{P}\left(Z_{tj} \leq -\sqrt{\frac{n(1+\sigma^2)}{\log m}}\right)$$

$$\leq mn\mathbb{P}\left(Z_{tj} \leq -\sqrt{\frac{n(1+\sigma^2)}{\log m}}\right) \tag{9}$$

$$\overset{(c)}{\leq} mn \exp\left(-\frac{n}{2\log m}\right) \tag{10}$$

where $(a)$ follows by the union bound, $(b)$ follows since $c_{tj} = Z_{tj} + \ell_{tj}$ and $0 \leq \ell_{tj} \leq 1$, and $(c)$ follows by using that for $Z \sim \mathcal{N}(0, \sigma^2)$ the complementary CDF $\mathbb{P}(Z \geq x) \leq \exp(-x^2/2\sigma^2)$. Thus, Theorem 1 implies that the strategy $\widehat{p}^{\mathrm{EW}}$ which sets $p_t = p^{\mathrm{EW}}(c^{t-1})$ with learning rate $\alpha = \sqrt{\frac{\log m}{n(1+\sigma^2)}}$ achieves regret

$$\mathrm{Reg}(\widehat{p}^{\mathrm{EW}}, \mathrm{AWGN}(\sigma), \ell^n) \leq 2\sqrt{(1+\sigma^2)n\log m} + o(n). \tag{11}$$

**Uniform noise.** For additive channels with uniform noise, the channel output $c_{tj} = \ell_{tj} + Z_{tj}$ where $Z_{tj} \sim \mathrm{Unif}[-\sigma, \sigma]$ (so that the noise variance is $\sigma^2/3$). Since we are interested in how the regret scales as $\sigma$ increases, it suffices to assume that $\sigma \geq 1$. Then, consider the following estimator $\widehat{\ell}_t$ (which is a function of $c_t$):

$$\widehat{\ell}_{tj} = \begin{cases} -\sigma + \frac{1}{2} & \text{if } -\sigma \leq c_{tj} < -\sigma + 1 \\ \frac{1}{2} & \text{if } -\sigma + 1 \leq c_{tj} \leq \sigma \\ \sigma + \frac{1}{2} & \text{if } \sigma < c_{tj} \leq \sigma + 1. \end{cases} \tag{12}$$

We observe that $\mathsf{E}[\widehat{\ell}_{tj}] = \ell_{tj}$ i.e. $\widehat{\ell}_t$ is unbiased and that the MSE for this estimator satisfies $\mathsf{E}[\widehat{\ell}_{tj} - \ell_{tj}]^2 \leq \sigma$ (full calculations relegated to Appendix C).

For choice of learning rate $\alpha = \sqrt{\frac{\log m}{n(1+\sigma)}}$, we note that $\alpha \widehat{\ell}_{tj} \geq -\sigma\alpha = -\sigma\sqrt{\frac{\log m}{n(1+\sigma)}} \geq -1$ for large enough $n$. Therefore, if we use the strategy $\widehat{p}^{\mathrm{EW}}$ with the unbiased estimator $\widehat{\ell}_t$ in (12), Theorem 1 yields

$$\mathrm{Reg}(\widehat{p}^{\mathrm{EW}}, \mathrm{Unif}(\sigma), \ell^n) \leq 2\sqrt{(1+\sigma)n\log m}. \tag{13}$$

In Section 5, we show a matching lower bound to (13) and establish that the regret must grow as $\Omega\left(\sqrt{(1+\sigma)n\log m}\right)$, showing the tightness of (13).

**Symmetric noise with tail constraints.** If the additive noise is symmetric, i.e. the distribution of noise $Z$ and $-Z$ is the same, the most natural unbiased estimator for $\ell_t$ is $\widehat{\ell}_t = c_t$ (since the noise is additive and $0-$mean) which achieves mean-square error $\mathsf{E}[c_{tj} - \ell_{tj}]^2 = \sigma^2$ where $\sigma^2$ is the variance of the noise $Z_{tj}$. In order to apply Theorem 1 with $\alpha = \sqrt{\frac{\log m}{n(1+\sigma^2)}}$ to achieve regret scaling as $O(\sqrt{(1+\sigma^2)n\log m})$ (as in the AWGN channel setting) we need to establish a bound on $\mathbb{P}(\mathcal{E})$. Following the line of reasoning employed to reach (9) we have

$$\mathbb{P}(\mathcal{E}) \leq mn\mathbb{P}\left(\frac{Z_{tj}}{\sigma} \leq -\sqrt{\frac{n(1+\sigma^2)}{\sigma^2 \log m}}\right) \leq mn\mathbb{P}\left(\frac{Z_{tj}}{\sigma} \leq -\sqrt{\frac{n}{\log m}}\right)$$

which implies that a noise density with polynomially decaying tails (in particular for $\sigma = 1$, if the random variable $Z$ satisfies for large $x$ that $\mathbb{P}(Z \geq x) \leq \frac{c}{x^{6+\epsilon}}$ where $c$ is a positive absolute constant and $\epsilon > 0$) suffices to achieve regret

$$2\sqrt{(1+\sigma^2)n\log m} + o(n). \tag{14}$$

An important class of distributions that achieves this tail condition is *log-concave* distributions [SW14], which are distributions having density $f(z)$ for which the function $z \mapsto \log f(z)$ is concave. This class has a special significance across statistics and information theory and includes distributions such as the Gaussian distribution, the uniform distribution and the Laplace distribution. Since all log-concave distributions are subexponential (i.e. have exponentially decaying tails) these satisfy aforementioned the condition on $\mathbb{P}(\mathcal{E})$ as $n$ grows larger. While for the specific cases of Gaussian and Laplace densities, it is possible to achieve a matching $\Omega(\sqrt{(1+\sigma^2)n\log m})$ lower bound for the regret, the most general lower bound we are able to achieve is a fundamental lower bound of $\Omega(\sqrt{(1+\sigma)n\log m})$ on the regret when the class of noise densities is log-concave. While it might appear that the bound can be strengthened in general, we have seen that this fundamental lower bound can in fact be achieved for uniform noise distributions by constructing a different unbiased estimator that achieves $O(\sqrt{(1+\sigma)n\log m})$ regret.

## 5 Lower bounds

In this section, we establish fundamental lower bounds on the regret $\max_{\ell^n} \text{Reg}(p, P_{c|\ell}, \ell^n)$ for any strategy $p$. To this end, we need the following definition.

**Definition 1** *The strong data processing constant of a binary-input channel $P_{Y|X}$ is defined as*

$$\eta(P_{Y|X}) = \sup_{P_X \neq Q_X} \frac{D(P_X \circ P_{Y|X} \| Q_X \circ P_{Y|X})}{D(P_X \| Q_X)} \tag{15}$$

*where $P_X$ and $Q_X$ are distributions defined on $\{0, 1\}$.*

Intuitively, this measure quantifies some sense of "loss of information" in a noisy channel—this interpretation is more clear by the alternate representation of $\eta(P_{Y|X})$ (see [PW22, Theorem 33.5])

$$\eta(P_{Y|X}) = \sup_{P_{UX}: U \to X \to Y} \frac{I(U;Y)}{I(U;X)} \tag{16}$$

where $U$ is an auxiliary random variable, and $U \to X \to Y$ represents a Markov chain. The data processing inequality [CT06] from information theory immediately implies that $\eta(P_{Y|X}) \leq 1$; often, as we show subsequently, we can establish $\eta(P_{Y|X}) < 1$. There has been much interest in characterizing $\eta(P_{Y|X})$ for various channels due to numerous applications arising in the domain of statistical inference—see [PW17, Rag16], [PW22, Chapter 33] for a detailed survey.

Next, we state Theorem 2 formally. This result is stated in [BK24] as Theorem 2 (without a full proof), and we provide a full proof in Appendix D.

**Theorem 2** *If the noise is memoryless and component-wise independent (i.e. $P_{c|\ell} = \prod_{j=1}^m \mathsf{P}_{c_j|\ell_j}$) then*

$$\sup_{\ell^n} \text{Reg}(p, P_{c|\ell}, \ell^n) \geq \sqrt{\frac{n\log(m/4)}{16\eta(\mathsf{P}_{c|\ell})}} \tag{17}$$

*where with some abuse of notation, $\eta(\mathsf{P}_{c|\ell})$ (as in Definition 1) restricts the channel to binary input $\{0, 1\}$.*

We now instantiate Theorem 2 for the class of additive noise channels, recalling that for these channels $c_{tj} = \ell_{tj} + Z_{tj}$ for (independent and identically distributed) random variables $Z_{tj}$. To quantify $\eta(\mathsf{P}_{c|\ell})$, we will utilize the following characterization from [PW17, Theorem 21]

**Theorem 3** *For a binary-input channel $P_{Y|X}$,*

$$\frac{H^2(P_{Y|X=0}, P_{Y|X=1})}{2} \leq \eta(P_{Y|X}) \leq H^2(P_{Y|X=0}, P_{Y|X=1}) \tag{18}$$

*where $H$ represents the Hellinger divergence between two distributions.*

We can now use this result for the specific noise models we are interested in.

**Additive white Gaussian noise.** If $Z_{tj} \sim \mathcal{N}(0, \sigma^2)$, then (18) implies that

$$
\eta(\text{AWGN}(\sigma^2)) = H^2(\mathcal{N}(0, \sigma^2), \mathcal{N}(1, \sigma^2))
$$

$$
= 1 - \frac{1}{2\pi\sigma^2} \int_{-\infty}^{\infty} \exp\left(-\frac{x^2}{2\sigma^2} - \frac{(x-1)^2}{2\sigma^2}\right) dx \tag{19}
$$

$$
= 2 - 2e^{-1/8\sigma^2} \leq \frac{4}{1 + \sigma^2} \tag{20}
$$

where (20) follows since $(1 - e^{-1/8x^2})(1 + x^2) \leq 4$ for all $x$. Using (20) in Theorem 2 implies that

$$
\sup_{\ell^n} \text{Reg}(p, \text{BEC}(\mathsf{e}), \ell^n) \geq \sqrt{\frac{(1 + \sigma^2)n \log(m/4)}{64}} \tag{21}
$$

matching up to constants the upper bound in (11).

**Additive uniform noise.** The uniform additive noise channel has the noise $Z_{tj} \sim \text{Unif}[-\sigma, \sigma]$—note that this noise has variance $\sigma^2/3$. In this case $P_{Y|X=0} = \text{Unif}[-\sigma, \sigma]$ with density $f_0(x) = \frac{1}{2\sigma} \mathbb{1}\{-\sigma \leq x \leq \sigma\}$, and $P_{Y|X=1} = \text{Unif}[-\sigma + 1, \sigma + 1]$ with density $f_1(x) = \frac{1}{2\sigma} \mathbb{1}\{-\sigma + 1 \leq x \leq \sigma + 1\}$. Let us assume that $\sigma \geq 1$; in this case

$$
\eta(\text{Unif}(\sigma)) \leq H^2(\text{Unif}[-\sigma, \sigma], \text{Unif}[-\sigma + 1, \sigma + 1])
$$

$$
= 1 - \frac{1}{2\sigma} \int_{-\sigma+1}^{\sigma+1} dx = \frac{1}{2\sigma}. \tag{22}
$$

Combining (22) with the trivial bound $\eta \leq 1$ yields

$$
\eta(\text{Unif}(\sigma)) \leq \frac{1}{\sigma + 1} \tag{23}
$$

for all $\sigma > 0$; implying the fundamental lower bound on the regret when the feedback is corrupted with additive uniform noise

$$
\sup_{\ell^n} \text{Reg}(p, \text{BEC}(\mathsf{e}), \ell^n) \geq \sqrt{\frac{(1 + \sigma)n \log(m/4)}{16}} \tag{24}
$$

matching the upper bound result obtained in (13) up to constants.

**Additive symmetric, log-concave noise.** So far, in the additive noise examples we have considered (Gaussian and uniform noise), we established that noisy feedback incurs a multiplicative cost (over the noiseless case) on the regret that depends on the moments of the noise and this cost is strictly greater than 1 ($\sqrt{1 + \sigma^2}$ and $\sqrt{1 + \sigma}$ respectively). In light of the upper bound result in (14), we might hope that for general additive noise channels with mild tail conditions on the noise one can achieve $\eta(P_{c|\ell}) \geq \Omega(\sigma)$. Unfortunately, this is not the case in general—consider the additive channel $Y = X + Z$ with noise distribution $Z \sim \text{Uniform}\{-\sigma, \sigma\}$—this noise distribution is bounded; but still $\eta(P_{Y|X}) = 1$ since given $Y$, $X$ is perfectly known. Therefore, to obtain a more general result, more conditions need to be imposed on the noise distribution.

We will show a lower bound for the general class of symmetric log-concave distributions considered in Section 4.1, which encompasses the Gaussian and uniform distributions considered previously. Consider a log-concave noise distribution with variance $\sigma^2$ and let $f$ denote its density. Then,

$$
\eta(P_{Y|X}) \leq H^2(P_{Y|X=0}, P_{Y|X=1})
$$

$$
\overset{(a)}{\leq} 2TV(P_{Y|X=0}, P_{Y|X=1})
$$

$$
\overset{(b)}{=} \int_{-\infty}^{\infty} |f(z) - f(z-1)| dz \tag{25}
$$

where $(a)$ follows from the well known inequality $H^2 \leq 2TV$ between Hellinger and total variation distances, and $(b)$ follows from the definition of the total variation distance (and, the fact that the density of $Y|X = 1$ is $f(z-1)$). Next, we further simplify (25) using the symmetry and unimodality

of $f$ (since any log-concave distribution is also unimodal). Since $f(z)$ is decreasing for $z \geq 0$ and $f(z-1)$ is increasing for $z \leq 1$, for any $z \leq \frac{1}{2}$, we have

$$f(z-1) \leq f(1/2) \leq f(z)$$

and similarly for $z > \frac{1}{2}$, $f(z) \leq f(z-1)$. Therefore,

$$
\begin{aligned}
\int_{-\infty}^{\infty} |f(z) - f(z-1)| dz &= \int_{-\infty}^{1/2} (f(z) - f(z-1)) dz + \int_{1/2}^{\infty} (f(z-1) - f(z)) dz \\
&= 2 \int_{1/2}^{\infty} (f(z-1) - f(z)) dz \qquad\qquad (26) \\
&= 2 \left( \int_{1/2}^{\infty} f(z-1) dz - \int_{1/2}^{\infty} f(z) dz \right) \\
&= 2 \left( \int_{-1/2}^{\infty} f(z) dz - \int_{1/2}^{\infty} f(z) dz \right) \\
&= 2 \left( \int_{-1/2}^{1/2} f(z) dz \right) \\
&\leq \frac{4}{\sigma} \qquad\qquad\qquad\qquad\qquad\qquad\qquad (27)
\end{aligned}
$$

where (27) is due to the following proposition, a proof of which is provided in Appendix E.

**Proposition 1** *For a symmetric, log-concave distribution with variance $\sigma^2$, its density satisfies $f(z) \leq \frac{2}{\sigma}$.*

Putting together (27) and (25) along with the trivial bound $\eta \leq 1$, we see that for any additive noise channel with a symmetric, log-concave density

$$\eta(P_{Y|X}) \leq \frac{8}{(1+\sigma)}. \qquad\qquad (28)$$

This furthermore implies in the experts problem that if feedback is available with additive noise $c_{tj} = \ell_{tj} + Z_{tj}$ where $Z_{tj}$ is symmetric and log-concave, then

$$\sup_{\ell^n} \text{Reg}(p, \text{BEC}(\mathsf{e}), \ell^n) \geq \sqrt{\frac{(1+\sigma)n \log(m/4)}{128}}. \qquad\qquad (29)$$

It is interesting to note that the lower bound in (29) is not tight in general. This is true in particular for Gaussian noise and Laplace (double exponential) additive noise—for both, we can establish a $\sqrt{1+\sigma^2}$ scaling by direct computation of $H^2(P_{Y|X=0}, P_{Y|X=1})$. Nonetheless, it is tight for uniform noise, which is a log-concave distribution, as we have shown a matching upper bound in (13). Thus, it is tight in the sense that it cannot be improved without imposed further restrictions on the class of noise densities.

## 6 Discussion

This paper provides a comprehensive framework for prediction with expert advice under additive noise feedback, establishing tight regret bounds for Gaussian, uniform and Laplace noise, and nearly-tight bounds for the broader class of symmetric log-concave distributions. Our analysis reveals important differences in how noise characteristics affect learning performance—with regret penalties scaling quadratically with standard deviation for Gaussian noise but only linearly for uniform noise. We identify the strong data-processing coefficient as a critical measure characterizing how channel degradation impacts regret bounds. Important future directions include developing high-probability guarantees beyond expected regret analysis for risk-sensitive applications, and extending our framework to alternative loss functions beyond linear losses and to infinite expert classes—connecting this work to broader statistical learning theory through concepts like Rademacher complexity under noisy observations.

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

# A Literature Review

A closely related area where sequential decision-making with noisy feedback has been considered is control. The question examined here is how control systems can maintain stability and performance despite the presence of noise in the feedback loop. While measurement-feedback control is a classical topic [KSH00], the line of work [TM04b, TM04a, TSM04, KH19] examines fundamental limits of control performance when the feedback is subject to communication constraints.

[RWH+12] considered sequential anomaly detection and sequential probability assignment (i.e. online prediction using the logarithmic loss [Ris84, XB97, BK21, CBL06]) in the presence of noise and established minmax regret guarantees. Also in the setting of sequential probability assignment, [SRV18] considered compressed side information available noncausally—our work considers compressed feedback available causally, in the prediction with experts setting. Decision-making with noisy feedback in the sequential classification setting has been considered in [BDPSS09, WGS23]. The effect of noisy observations on the equlibrium value of games was characterized in [HACM22, SPJM23]. The setting where rather than the feedback $\ell_t$ the action $p_t$ is communicated over a noisy channel is considered in [DRS15, PGZ22, HKYF23], and minmax bounds on the regret incurred are established. The line of work [ACT20a, ACT20b, ACL+21] considers sequential statistical inference under constraints, designing optimal policies and well as establishing fundamental converse bounds.

We remark that our setting is distinct from that of i.i.d. $\ell_t$ and adversarially injected corruptions [AAK+20], a model which aims to bridge the distance between the case where the losses $\ell_t$ at chosen i.i.d. and the individual-sequence case (adversarial $\ell_t$). Moreover, our choice of benchmark being $\min_{j \in [m]} \sum_{t=1}^{n} \ell_{tj}$ (see the regret definition (3)) makes our setting distinct from smoothed analysis [HRS20, BHS23], where the benchmark is the best expert in hindsight on the noisy loss function—making smoothed analysis a beyond-worst case setting.

# B Proof of Theorem 1

First, the following proposition justifies the use of an unbiased estimator in the strategy.

**Proposition 2** *Let $\widehat{\ell}_t$ (where $\widehat{\ell}_t$ is a possibly noisy function of $c^t$) be such that $\mathsf{E}[\widehat{\ell}_t | \widehat{\ell}^{t-1}] = \ell_t$, and $p$ be any strategy for the noiseless experts problem. Then, the strategy $\widehat{p}$ that plays $\widehat{p}_t = p_t(\widehat{\ell}^{t-1})$ achieves*

$$\operatorname{Reg}(p, P_{c|\ell}, \ell^n) \leq \mathsf{E}\left[\sum_{t=1}^{n} \langle p(\widehat{\ell}^{t-1}), \widehat{\ell}_t \rangle - \min_{j \in [m]} \sum_{t=1}^{n} \widehat{\ell}_{tj}\right].$$

**Proof.**

$$
\begin{aligned}
\langle \mathsf{E}[p_t(\widehat{\ell}^{t-1})], \ell_t \rangle &= \mathsf{E}[\langle p_t(\widehat{\ell}^{t-1}), \ell_t \rangle] \\
&\stackrel{(a)}{=} \mathsf{E}[\langle p_t(\widehat{\ell}^{t-1}), \mathsf{E}[\widehat{\ell}_t | \widehat{\ell}^{t-1}] \rangle] \\
&= \mathsf{E}[\mathsf{E}[\langle p_t(\widehat{\ell}^{t-1}), \widehat{\ell}_t \rangle | \widehat{\ell}^{t-1}]] \\
&\stackrel{(c)}{=} \mathsf{E}\left[\langle p_t(\widehat{\ell}^{t-1}), \widehat{\ell}_t \rangle\right]
\end{aligned}
\tag{30}
$$

where $(a)$ follows by the conditional unbiasedness of $\widehat{\ell}_t$ and $(b)$ follows by the tower property of expectation. Moreover,

$$\min_{j \in [m]} \sum_{t=1}^{n} \ell_{tj} \stackrel{(a)}{=} \min_{j \in [m]} \mathsf{E}\left[\sum_{t=1}^{n} \widehat{\ell}_{tj}\right] \stackrel{(b)}{\geq} \mathsf{E}\left[\min_{j \in [m]} \sum_{t=1}^{n} \widehat{\ell}_{tj}\right] \tag{31}$$

where $(a)$ follows by the unbiasedness of $\widehat{\ell}_t$ and linearity of expectation, and $(b)$ follows since $\mathsf{E}[\min(\cdot)] \leq \min \mathsf{E}[\cdot]$. The Proposition follows by summing up (30) over $t$ and from (31). ∎

Proposition 2 establishes that upon construction of an unbiased estimator $\widehat{\ell}_t$, the decision-maker can pretend that the benchmark is $\min_{j \in [m]} \sum_{t=1}^{n} \widehat{\ell}_{tj}$, and employ a no-regret strategy for this benchmark.

To construct a scheme, we need to utilize a no-regret strategy for the noiseless setting in conjunction with an unbiased estimator $\widehat{\ell}_t$. To this end, we utilize the landmark exponential weights/Hedge (EW) strategy. We will need the following analysis of the exponential weights strategy $p^{\mathrm{EW}}$ (see for example [Luo22]), which bounds the regret incurred by $p^{\mathrm{EW}}$ in terms of the second moment of the loss functions [CBMS07, GSVE14].

**Lemma 1** *If $p_{tj}^{\mathrm{EW}}$ is chosen to be the exponential weights strategy, and if $\ell^n$ and $\alpha$ satisfy $-\alpha\ell_{tj} \le 1$ for all $t$ and $j$, we have*

$$\mathrm{Reg}(p^{\mathrm{EW}}, \ell^n) \le \frac{\log m}{\alpha} + \alpha \sum_{t=1}^{n} \sum_{j=1}^{m} p_{tj}^{\mathrm{EW}} \ell_{tj}^2. \tag{32}$$

**Proof.** Define

$$Z_t := \sum_{j=1}^{m} \exp\left(-\alpha \sum_{i=1}^{t-1} \ell_{ij}\right),$$

the normalizing term in $p_t^{\mathrm{EW}}$, so that $p_{tj}^{\mathrm{EW}} = \exp\left(-\alpha \sum_{i=1}^{t-1} \ell_{ij}\right)/Z_t$. Then, we will consider $\log Z_t$ to be the potential function and bound the difference in the potential function at each step. We have Note that

$$\begin{aligned}
\log Z_{t+1} - \log Z_t &= \log \frac{\sum_{j=1}^{m} \exp\left(-\alpha \sum_{i=1}^{t} \ell_{ij}\right)}{\sum_{j=1}^{m} \exp\left(-\alpha \sum_{i=1}^{t-1} \ell_{ij}\right)} \\
&= \log \frac{\sum_{j=1}^{m} \exp\left(-\alpha \sum_{i=1}^{t-1} \ell_{ij}\right) \exp(-\alpha\ell_{tj})}{\sum_{j=1}^{m} \exp\left(-\alpha \sum_{i=1}^{t-1} \ell_{ij}\right)} \\
&\overset{(a)}{=} \log \left(\sum_{j=1}^{m} p_{tj}^{\mathrm{EW}} \exp(-\alpha\ell_{tj})\right) \\
&\overset{(b)}{\le} \log \left(\sum_{j=1}^{m} p_{tj}^{\mathrm{EW}} \left(1 - \alpha\ell_{tj} + \alpha^2\ell_{tj}^2\right)\right) \\
&= \log \left(1 - \alpha \sum_{j=1}^{m} p_{tj}^{\mathrm{EW}} \ell_{tj} + \alpha^2 \sum_{j=1}^{m} p_{tj} \ell_{tj}^2\right) \\
&\overset{(c)}{\le} -\alpha \sum_{j=1}^{m} p_{tj}^{\mathrm{EW}} \ell_{tj} + \alpha^2 \sum_{j=1}^{m} p_{tj}^{\mathrm{EW}} \ell_{tj}^2 \\
&= -\alpha \langle p_t^{\mathrm{EW}}, \ell_t \rangle + \alpha^2 \sum_{j=1}^{m} p_{tj}^{\mathrm{EW}} \ell_{tj}^2 \tag{33}
\end{aligned}$$

where $(a)$ follows by the definition of $p^{\mathrm{EW}}$, $(b)$ follows since $e^x \le 1 + x + x^2$ for $x \le 1$ (and $-\alpha\ell_{tj} \le 1$), and $(c)$ follows since $\log(1+x) \le x$ for all $x$. Now, we observe that

$$\begin{aligned}
\log Z_{n+1} &= \log \left(\sum_{j=1}^{m} \exp\left(-\alpha \sum_{t=1}^{n} \ell_{tj}\right)\right) \\
&\ge \max_{j \in [m]} \log \left(\exp\left(-\alpha \sum_{t=1}^{n} \ell_{tj}\right)\right) \\
&= -\alpha \min_{j \in [m]} \sum_{t=1}^{n} \ell_{tj} \tag{34}
\end{aligned}$$

and that $Z_1 = m$. Summing up (33) over all $t \in [n]$, using (34) and rearranging yields the Lemma. ∎

Recall the achievability strategy $\widehat{p}^{\mathrm{EW}}$:

- Construct an unbiased estimator $\widehat{\ell}_t$ for $\ell_t$ from the channel output $c_t$.
- Play $p_t^{\text{EW}}(\widehat{\ell}^{t-1})$.

Define the "bad" event $\mathcal{E} := \{\exists t, j : -\alpha\widehat{\ell}_{tj} > 1\}$, which by the condition stated in the Theorem occurs with probability $\mathbb{P}(\mathcal{E})$. We will split the regret analysis into two cases: if $\mathcal{E}^C$ occurs, where Lemma 1 can be invoked, and if $\mathcal{E}$ occurs, where we will utilize a worst-case bound on regret. First, we use Proposition 2 which yields

$$\text{Reg}(\widehat{p}^{\text{EW}}, P_{c|\ell}, \ell^n) \leq \mathsf{E}\left[\sum_{t=1}^{n}\langle p^{\text{EW}}(\widehat{\ell}^{t-1}), \widehat{\ell}_t\rangle - \min_{j\in[m]}\sum_{t=1}^{n}\widehat{\ell}_{tj}\right] \tag{35}$$

and we have

$$\mathsf{E}\left[\sum_{t=1}^{n}\langle p^{\text{EW}}(\widehat{\ell}^{t-1}), \widehat{\ell}_t\rangle - \min_{j\in[m]}\sum_{t=1}^{n}\widehat{\ell}_{tj}\right]$$

$$= \mathsf{E}\left[\left(\sum_{t=1}^{n}\langle p^{\text{EW}}(\widehat{\ell}^{t-1}), \widehat{\ell}_t\rangle - \min_{j\in[m]}\sum_{t=1}^{n}\widehat{\ell}_{tj}\right)\mathbb{1}\{\mathcal{E}^C\}\right]$$

$$+ \mathsf{E}\left[\sum_{t=1}^{n}\langle p^{\text{EW}}(\widehat{\ell}^{t-1}), \widehat{\ell}_t\rangle\mathbb{1}\{\mathcal{E}\}\right] - \mathsf{E}\left[\left(\min_{j\in[m]}\sum_{t=1}^{n}\widehat{\ell}_{tj}\right)\mathbb{1}\{\mathcal{E}\}\right]. \tag{36}$$

We analyze the three terms in the right hand side of (36) separately. First, note that if $\mathcal{E}^C$ occurs, then the conditions in Lemma 1 are satisfied which can be employed to get

$$\left(\sum_{t=1}^{n}\langle p^{\text{EW}}(\widehat{\ell}^{t-1}), \widehat{\ell}_t\rangle - \min_{j\in[m]}\sum_{t=1}^{n}\widehat{\ell}_{tj}\right)\mathbb{1}\{\mathcal{E}^C\} \leq \frac{\log m}{\alpha} + \alpha\sum_{t=1}^{n}\sum_{j=1}^{m}p_{tj}^{\text{EW}}(\widehat{\ell}^{t-1})\widehat{\ell}_{tj}^2 \tag{37}$$

where (37) also uses that indicator is bounded by 1 and the term to be multiplied is positive. Next, note that

$$\mathsf{E}[p_{tj}^{\text{EW}}(\widehat{\ell}^{t-1})\widehat{\ell}_{tj}^2] = \mathsf{E}[p_{tj}^{\text{EW}}(\widehat{\ell}^{t-1})\ell_{tj}^2] + \mathsf{E}[p_{tj}^{\text{EW}}(\widehat{\ell}^{t-1})(\widehat{\ell}_{tj} - \ell_{tj})^2] + \mathsf{E}[2p_{tj}^{\text{EW}}(\widehat{\ell}^{t-1})\ell_{tj}(\widehat{\ell}_{tj} - \ell_{tj})]$$

$$\overset{(a)}{=} \mathsf{E}[p_{tj}^{\text{EW}}(\widehat{\ell}^{t-1})\ell_{tj}^2] + \mathsf{E}[p_{tj}^{\text{EW}}(\widehat{\ell}^{t-1})]\mathsf{E}[(\widehat{\ell}_{tj} - \ell_{tj})^2]$$

$$\overset{(b)}{\leq} \mathsf{E}[p_{tj}^{\text{EW}}(\widehat{\ell}^{t-1})]\left(1 + \max_{t,j}\mathsf{E}[(\widehat{\ell}_{tj} - \ell_{tj})^2]\right) \tag{38}$$

where $(a)$ follows from the fact that $\widehat{\ell}_t$ is independent of $\widehat{\ell}^{t-1}$ and that $\widehat{\ell}_t$ is unbiased, and $(b)$ uses that $\ell_{tj}^2 \leq 1$ by assumption. Taking expectations on both sides of (37) and (38) yields

$$\mathsf{E}\left[\left(\sum_{t=1}^{n}\langle p^{\text{EW}}(\widehat{\ell}^{t-1}), \widehat{\ell}_t\rangle - \min_{j\in[m]}\sum_{t=1}^{n}\widehat{\ell}_{tj}\right)\mathbb{1}\{\mathcal{E}^C\}\right]$$

$$\leq \frac{\log m}{\alpha} + \alpha\sum_{t=1}^{n}\left(1 + \max_{t,j}\mathsf{E}[(\widehat{\ell}_{tj} - \ell_{tj})^2]\right)\mathsf{E}\left[\sum_{j=1}^{m}p_{tj}^{\text{EW}}(\widehat{\ell}^{t-1})\right]$$

$$= \frac{\log m}{\alpha} + \alpha n\left(1 + \max_{t,j}\mathsf{E}[(\widehat{\ell}_{tj} - \ell_{tj})^2]\right). \tag{39}$$

To bound the second term in (36), we apply

$$\sum_{t=1}^{n}\mathsf{E}\left[\langle p^{\text{EW}}(\widehat{\ell}^{t-1}), \widehat{\ell}_t\rangle\mathbb{1}\{\mathcal{E}\}\right] \leq \sum_{t=1}^{n}\mathsf{E}\left[|\langle p^{\text{EW}}(\widehat{\ell}^{t-1}), \widehat{\ell}_t\rangle|\mathbb{1}\{\mathcal{E}\}\right]$$

$$\overset{(a)}{\leq} \sum_{t=1}^{n}\mathsf{E}\left[\|\widehat{\ell}_t\|_\infty\mathbb{1}\{\mathcal{E}\}\right]$$

$$\overset{(b)}{\leq} \sum_{t=1}^{n}\mathsf{E}\left[\sum_{j=1}^{m}|\widehat{\ell}_{tj}|\mathbb{1}\{\mathcal{E}\}\right]$$

$$\overset{(c)}{\leq} \sum_{t=1}^{n} \sum_{j=1}^{m} \sqrt{\mathsf{E}[\widehat{\ell}_{tj}]^2} \sqrt{\mathbb{P}(\mathcal{E})}$$

$$\overset{(d)}{\leq} mn\sqrt{\left(1 + \max_{t,j} \mathsf{E}[\widehat{\ell}_{tj} - \ell_{tj}]^2\right) \mathbb{P}(\mathcal{E})} \tag{40}$$

where $(a)$ uses the Holder inequality and the fact that $p^{\mathrm{EW}}(c^{t-1})$ is a probability distribution, $(b)$ uses the fact that the absolute maximum in a vector is bounded by the sum of the absolute values, $(c)$ uses the Cauchy–Schwartz inequality, and $(d)$ uses unbiasedness of $\widehat{\ell}_{tj}$ along with the fact that $\ell_{tj}^2 \leq 1$. The third term in (36) will be dealt with similarly:

$$\mathsf{E}\left[\left(-\min_j \sum_{t=1}^{n} \widehat{\ell}_{tj}\right) \mathbb{1}\{\mathcal{E}\}\right] \leq \sum_{j=1}^{m} \sum_{t=1}^{n} \mathsf{E}\left[|\widehat{\ell}_{tj}|\mathbb{1}\{\mathcal{E}\}\right]$$

$$\leq mn\sqrt{\left(1 + \max_{t,j} \mathsf{E}[\widehat{\ell}_{tj} - \ell_{tj}]^2\right) \mathbb{P}(\mathcal{E})} \tag{41}$$

where (41) follows from (40). Finally, using (39), (40) and (41) in (36) concludes the proof.

## C  Upper bound for uniform additive noise

We first show that the estimator $\widehat{\ell}_{tj}$ in (12) is unbiased. Note that

$$\mathsf{E}[\widehat{\ell}_t] = \mathsf{E}\left[\left(-\sigma + \frac{1}{2}\right) \mathbb{1}\{-\sigma \leq c_{tj} < -\sigma + 1\} + \frac{1}{2}\mathbb{1}\{-\sigma + 1 \leq c_{tj} < \sigma\}\right.$$

$$\left. + \left(\sigma + \frac{1}{2}\right) \mathbb{1}\{\sigma \leq c_{tj} \leq \sigma + 1\}\right]$$

$$= \left(-\sigma + \frac{1}{2}\right) \mathbb{P}\left(-\sigma \leq c_{tj} < -\sigma + 1\right) + \frac{1}{2}\mathbb{P}\left(-\sigma + 1 \leq c_{tj} < \sigma\right)$$

$$+ \left(\sigma + \frac{1}{2}\right) \mathbb{P}\left(\sigma \leq c_{tj} < \sigma + 1\right) \tag{42}$$

Since $c_{tj} = \ell_{tj} + Z_{tj}$ is distributed as $\mathrm{Unif}[-\sigma + \ell_{tj}, \sigma + \ell_{tj}]$, we have

$$\mathbb{P}\left(-\sigma \leq c_{tj} < -\sigma + 1\right) = \frac{1 - \ell_{tj}}{2\sigma} \tag{43}$$

$$\mathbb{P}\left(-\sigma + 1 \leq c_{tj} < \sigma\right) = \frac{2\sigma - 1}{2\sigma} \tag{44}$$

$$\mathbb{P}\left(\sigma \leq c_{tj} \leq \sigma + 1\right) = \frac{\ell_{tj}}{2\sigma}. \tag{45}$$

Substituting (43), (44) and (45) in (42) yields

$$\mathsf{E}[\widehat{\ell}_t] = \frac{(-2\sigma + 1)(1 - \ell_{tj})}{4\sigma} + \frac{2\sigma - 1}{4\sigma} + \frac{(2\sigma + 1)\ell_{tj}}{4\sigma}$$

$$= \ell_{tj} \tag{46}$$

The MSE for this estimator satisfies

$$\mathsf{E}[\widehat{\ell}_{tj} - \ell_{tj}]^2 = \mathsf{E}[\widehat{\ell}_{tj}^2] - \ell_{tj}^2$$

$$= \left(-\sigma + \frac{1}{2}\right)^2 \mathbb{P}\left(-\sigma \leq c_{tj} < -\sigma + 1\right) + \frac{1}{4}\mathbb{P}\left(-\sigma + 1 \leq c_{tj} < \sigma\right)$$

$$+ \left(\sigma + \frac{1}{2}\right)^2 \mathbb{P}\left(\sigma \leq c_{tj} < \sigma + 1\right) - \ell_{tj}^2$$

$$\overset{(a)}{=} \frac{(2\sigma - 1)^2(1 - \ell_{tj})}{8\sigma} + \frac{(2\sigma - 1)}{8\sigma} + \frac{(2\sigma + 1)^2 \ell_{tj}}{8\sigma} - \ell_{tj}^2$$

$$= \frac{\sigma}{2} - \left(\ell_{tj} - \frac{1}{2}\right)^2$$

$$\leq \sigma \tag{47}$$

where $(a)$ uses (43), (44) and (45).

## D   Proof of Theorem 2

Consider the following (random) ensemble of loss vectors:

- Pick $J^* \sim \mathrm{Uniform}[m]$.
- Given $J^* = j^*$, the loss vectors $\ell^n$ are generated i.i.d., with independent components as per the distribution

$$\ell_{tj} \sim \begin{cases} \mathrm{Bern}(1/2 - \epsilon), & \text{if } j = j^* \\ \mathrm{Bern}(1/2), & \text{otherwise} \end{cases} \tag{48}$$

for some $0 < \epsilon < 1/4$ to be determined later.

Intuitively, in order to achieve sublinear regret in $n$ with these loss functions, the decision-maker must eventually detect the expert $j^*$ that has the lowest bias and therefore this can be thought of as a hypothesis testing problem. To formalize this, we have

$$\sup_{\widetilde{\ell}^n} \mathrm{Reg}(p, P_{c|\ell}, \widetilde{\ell}^n) \geq \mathsf{E}\left[\sum_{t=1}^{n} \langle p_t(c^{t-1}), \ell_t \rangle\right] - \mathsf{E}\left[\min_{j \in [m]} \sum_{t=1}^{n} \ell_{tj}\right]. \tag{49}$$

Now, note that

$$\mathsf{E}\left[\min_{j \in [m]} \sum_{t=1}^{n} \ell_{tj}\right] \overset{(a)}{\leq} \mathsf{E}\left[\min_{j \in [m]} \mathsf{E}\left[\sum_{t=1}^{n} \ell_{tj} \,\middle|\, J^*\right]\right] \tag{50}$$

$$\overset{(b)}{=} n\left(\frac{1}{2} - \epsilon\right) \tag{51}$$

where $(a)$ follows since $\mathsf{E}[\min(\cdot)] \leq \min \mathsf{E}[\cdot]$ and $(b)$ follows since by the distribution on the losses in (48)

$$\mathsf{E}[\ell_{tj}|J^*] = \begin{cases} \frac{1}{2}, & \text{if } j = J^* \\ \frac{1}{2} - \epsilon. & \text{otherwise} \end{cases} \tag{52}$$

To further bring out the analogy between hypothesis testing and the regret, we note that for the random variable distributed as $J_t \sim p_t(c^{t-1})$ conditional on $c^{t-1}$ (i.e. a random expert is chosen as per the distribution $p_t(c^{t-1})$)

$$\mathsf{E}[\langle p_t(c^{t-1}), \ell_t \rangle | c^{t-1}, \ell_t] = \mathsf{E}[\ell_{tJ_t} | c^{t-1}, \ell_t],$$

and therefore

$$\mathsf{E}[\langle p_t(c^{t-1}), \ell_t \rangle] = \mathsf{E}[\ell_{tJ_t}].$$

Then,

$$\mathsf{E}[\langle p_t(c^{t-1}), \ell_t \rangle] = \mathsf{E}[\mathsf{E}[\ell_{tJ_t} | J_t]]$$

$$\overset{(a)}{=} \mathsf{E}\left[\frac{1}{2} \mathbb{1}\{J_t \neq J^*\} + \left(\frac{1}{2} - \epsilon\right) \mathbb{1}\{J_t = J^*\}\right]$$

$$= \frac{1}{2} - \epsilon \mathbb{P}[J_t = J^*] \tag{53}$$

where $(a)$ follows from (52). Using (53) along with (51) and (49) yields

$$\sup_{\widetilde{\ell}^n} \mathrm{Reg}(p, P_{c|\ell}, \widetilde{\ell}^n) \geq \epsilon \sum_{t=1}^{n} \mathbb{P}[J_t \neq J^*]. \tag{54}$$

To further lower bound the regret, we apply the Fano inequality to each term in the right hand side of (54)

$$\mathbb{P}[J_t(c^{t-1}) \neq J^*] \geq 1 - \frac{I(J^*; J_t) + \log 2}{\log m} \tag{55}$$

$$\geq 1 - \frac{I(J^*; c^{t-1}) + \log 2}{\log m}, \tag{56}$$

where (56) follows by the data processing inequality since $J^* \to c^{t-1} \to J_t$.

Since the noise is memoryless by assumption,

$$
\begin{aligned}
I(J^*; c^t) &= H(c^t) - H(c^t | J^*) \\
&\overset{(a)}{\leq} \sum_{i=1}^{t} H(c_i) - H(c^t | J^*) \\
&\overset{(b)}{=} \sum_{i=1}^{t} \left( H(c_i) - H(c_i | J^*) \right) \\
&= \sum_{i=1}^{t} I(J^*; c_i) \\
&\overset{(c)}{\leq} t I(J^*; c_1).
\end{aligned}
\tag{57}
$$

where $(a)$ follows by the subadditivity of entropy, $(b)$ follows since given $J^*$, $c^t$ are independent (because given $J^*$, $\ell^t$ are independent as per (48) and the channel is memoryless by assumption), and finally $(c)$ follows by symmetry ($l^t$ are identically distributed, therefore so are $c^t$). Next, we have

$$
\begin{aligned}
I(J^*; c_1) &= D\left( P_{c_1 | J^*} \| P_{c_1} \big| P_{J^*} \right) \\
&= \frac{1}{m} \sum_{j=1}^{m} D\left( P_{c_1 | J^*=j} \| P_{c_1} \right) \\
&= \frac{1}{m} \sum_{j=1}^{m} D\left( P_{c_1 | J^*=j} \Big\| \frac{1}{m} \sum_{j'=1}^{m} P_{c_1 | J^*=j'} \right) \\
&\overset{(a)}{\leq} \frac{1}{m^2} \sum_{j=1}^{m} \sum_{j'=1}^{m} D\left( P_{c_1 | J^*=j} \| P_{c_1 | J^*=j'} \right) \\
&\overset{(b)}{=} \frac{m^2 - m}{m^2} D\left( P_{c_1 | J^*=1} \| P_{c_1 | J^*=2} \right) \\
&\leq D\left( P_{c_1 | J^*=1} \| P_{c_1 | J^*=2} \right) \\
&\overset{(c)}{=} \sum_{j=1}^{m} D(P_{c_{1j} | J^*=1} \| P_{c_{1j} | J^*=2}) \\
&\overset{(d)}{=} D(P_{c_{11} | J^*=1} \| P_{c_{12} | J^*=2}) + D(P_{c_{12} | J^*=1} \| P_{c_{12} | J^*=2}) \\
&= D\left( \mathrm{Bern}(1/2 - \epsilon) \circ \mathsf{P}_{\mathsf{c}|\ell} \| \mathrm{Bern}(1/2) \circ \mathsf{P}_{\mathsf{c}|\ell} \right) \\
&\quad + D\left( \mathrm{Bern}(1/2) \circ \mathsf{P}_{\mathsf{c}|\ell} \| \mathrm{Bern}(1/2 - \epsilon) \circ \mathsf{P}_{\mathsf{c}|\ell} \right)
\end{aligned}
\tag{58}
$$

where $(a)$ follows since $D(P\|Q)$ is convex in the pair $P$ and $Q$, $(b)$ follows by symmetry, $(c)$ follows since the vector $c_1$ has a product distribution given $J^*$ (because $\ell_1$ has a product distribution and the noise is component-wise independent) and $(d)$ follows since all the other components except the first and second have the same distribution ($\mathrm{Bern}(1/2) \circ \mathsf{P}_{\mathsf{c}|\ell}$). Recalling the definition of $\eta(\mathsf{P}_{\mathsf{c}|\ell})$ in Definition 1, we have

$$D\left( \mathrm{Bern}(1/2 - \epsilon) \circ \mathsf{P}_{\mathsf{c}|\ell} \| \mathrm{Bern}(1/2) \circ \mathsf{P}_{\mathsf{c}|\ell} \right) \leq \eta(\mathsf{P}_{\mathsf{c}|\ell}) \left( d\left( \frac{1}{2} - \epsilon \Big\| \frac{1}{2} \right) \right)$$

$$\leq \eta(\mathsf{P}_{\mathsf{c}|\ell})\epsilon^2 \tag{59}$$

where $d(\cdot\|\cdot)$ denotes the binary KL divergence, and the final inequality follows since $\frac{d\left(\frac{1}{2}-x\|\frac{1}{2}\right)}{x^2} \leq 1$ for $x < 1/4$ and $\epsilon < 1/4$ by assumption. Using the same reasoning for the second term of (58), and using (59) in (57) we have

$$I(J^*; c^t) \leq 2t\eta(\mathsf{P}_{\mathsf{c}|\ell})\epsilon^2$$

and therefore from (54) and (56) we get

$$\sup_{\widetilde{\ell}^n} \mathrm{Reg}(p, P_{c|\ell}, \widetilde{\ell}^n) \geq \epsilon \sum_{t=1}^{n} \left( 1 - \frac{2(t-1)\eta(\mathsf{P}_{\mathsf{c}|\ell})\epsilon^2 + \log 2}{\log m} \right)$$

$$\geq n\epsilon \left( 1 - \frac{2n\eta(\mathsf{P}_{\mathsf{c}|\ell})\epsilon^2 + \log 2}{\log m} \right). \tag{60}$$

Finally, the choice of $\epsilon = \sqrt{\frac{\log(m/4)}{4n\eta(\mathsf{P}_{\mathsf{c}|\ell})}}$ (which guarantees $\epsilon \leq 1/4$ for a large enough $n$) in (60) yields

$$\sup_{\widetilde{\ell}^n} \mathrm{Reg}(p, P_{c|\ell}, \widetilde{\ell}^n) \geq \sqrt{\frac{n \log(m/4)}{16\eta(\mathsf{P}_{\mathsf{c}|\ell})}} \tag{61}$$

as claimed.

**Remark 1 (Lower bound for the noiseless problem)** *From (56), and since $J^* \to \ell^t \to c^t$, we see that $I(J^*; c^t) \leq I(J^*; \ell^t)$. Following the single-letterization argument in (57) and using the arguments leading up to (58) we can recover the the lower bound for the noiseless prediction with experts problem.*

## E    Proof of Proposition 1

Define

$$g(z) := f(0) \exp(-2xf(0)).$$

Then, $f(0) = g(0)$ and $\int_0^\infty (f(z) - g(z))dz = \int_0^\infty f(z)dz - \int_0^\infty g(z)dz = \frac{1}{2} - \frac{1}{2} = 0$. Since $f, g \to 0$ and $z \to \infty$, this implies that the function $f(z) - g(z)$ crosses the origin at least once in $z > 0$. Moreover, any solution of $f(z) - g(z) = 0 \implies f(z) = g(z)$ must satisfy also $\log f(z) - \log g(z) = 0$. Since $z \mapsto \log f(z) - \log g(z)$ is a concave function (by virtue of $f(z)$ being log-concave and $g(z)$ being log-affine), this implies that $\log f(z) - \log g(z)$ crosses the origin at most once in $z > 0$. Therefore, putting the two together implies that $f(z) - g(z) = 0$ occurs exactly at one point in $0 < z < \infty$. Let us call this point $t$, so that $f(t) = g(t)$. Therefore, for all $z \leq t$, $f(z) \geq g(z)$ and for all $z > t$, $f(z) \leq g(z)$. Putting these two together, we have

$$(f(z) - g(z))(t^2 - z^2) \geq 0$$

which implies that

$$\int_0^\infty z^2 f(z)dz \leq \int_0^\infty z^2 g(z)dz \tag{62}$$

Since $\int_0^\infty z^2 f(z)dz = \frac{\sigma^2}{2}$ and

$$\int_0^\infty z^2 g(z)dz = \int_0^\infty z^2 \exp(-2zf(0))dz = \frac{1}{8f(0)^2} \int_0^\infty z^2 \exp(-z)dz = \frac{1}{4f(0)^2},$$

(62) yields

$$f(0)^2 \leq \frac{1}{2\sigma^2} \tag{63}$$

which leads to the required Proposition.

