# OpenReview forum: "Prediction with expert advice under additive noise"
_NeurIPS.cc/2025/Conference — NeurIPS 2025 poster_

### Official Review · Reviewer_Tk3H · 2025-06-12

**Clarity:** 4
**Significance:** 3
**Originality:** 3
**Rating:** 5
**Confidence:** 4

**Summary:**

The paper considers the problem of prediction with expert advice with corrupted feedback, assuming that the learner only gets to observe the loss functions up to additive noise. Various upper bounds are given for specific noise models, which are supported by lower bounds that are shown to be tight for some choices of the noise distribution.

**Questions:**

N/A

**Ethical Concerns:**

["NO or VERY MINOR ethics concerns only"]

**Final Justification:**

I continue to support acceptance of this paper.

**Limitations:**

yes

**Quality:**

4

**Strengths And Weaknesses:**

I enjoyed reading the paper and found some of the results to be quite neat. In particular, I found the loss estimator uniform noise to be quite clever, and the resulting regret bound to be surprisingly tight. Together with the lower bounds, these results have revealed quite a bit of unexpected structure in this (otherwise quite innocent-looking) problem setting. On the technical side, I appreciated the clear and intuitive nature of both the upper bounds in Theorem 1 and the lower bounds in Theorem 2.

Playing the devil's advocate for a second, I can see that some readers may feel that the results are a bit "elementary". In particular, I myself have felt this to an extent as I started to read the paper and got to the upper bounds for the setting of Gaussian observation noise --- these results are not surprising at all. Similarly, Theorem 1 is far from being novel, and can be proved via very small adjustments to the classic analysis of exponential weighted averaging. Thankfully, the authors are reasonably low-key about this result and are not trying to make it look much more significant than it actually is.

Otherwise, the surprise factor for the case of uniform noise has saved this paper for me. Given the simplicity of the estimator and its error analysis, I suspect that this trick might have been known previously but I have no knowledge of this. If the authors have a pointer, I would appreciate that (and also maintain my positive rating since the trick is definitely new in the context of online learning either way).

Overall, I am happy to recommend acceptance.

---

> ### Author Rebuttal · Authors · 2025-07-30
>
> We thank the reviewer for their detailed and encouraging remarks.
> We have not seen the construction of the unbiased estimator for the uniform distribution appear previously in the literature. Since uniform error is closely related to noise due to quantization, we expect the tight results for the uniform error to translate into tight bounds for regret analysis with quantized feedback.

---

> > ### Comment · Reviewer_Tk3H · 2025-08-05
> >
> > Thank you for your response! I continue to support acceptance of this paper.

---

### Official Review · Reviewer_vFEj · 2025-06-29

**Clarity:** 3
**Significance:** 2
**Originality:** 3
**Rating:** 5
**Confidence:** 4

**Summary:**

This work investigates the setting of prediction with expert advice where each component $\ell_t(i)$ of the loss vector observed by the learner is affected by zero-mean additive iid noise (where the noise $Z_t(i)$ in each observed loss component $\ell_t(i) + Z_t(i)$ is iid across time $t$ and across experts $i$). As the noise may violate the requirement that $\alpha\ell_t(i) \ge -1$, where $\alpha$ is the learning rate, the regret bound now depends on the probability that in at least one round $t$ the boundedness loss condition $\alpha(\ell_t(i) + Z_t(i)) \ge -1$ is violated for some expert $i$. The authors prove upper and lower bounds on the regret when $Z_t(i)$ are symmetric log-concave distributions parameterized by their variance. While these bounds are not matching, the authors also prove that the upper bound is achieved by Gaussian distributions and the lower bound is achieved by the Uniform distribution (which are both log-concave).

**Questions:**

1. What additional properties of symmetric log-concave noise distributions could capture the right scaling of the regret?

2. Step (c) in display (40) seems to use independence between $\widehat{\ell}_{t,j}$ and $\mathcal{E}$, which appears to be invalid. Can you commment?

3. The analysis heavily relies on componentwise independence of the noise. Would the analysis of joint noise distributions require a significantly different approach?

4. While the upper bound depends on the variance of the MSE, the lower bound depends on an information-theoretic quantity. Can you comment on the feasibiliy of deriving upper bounds depending on similar information-theoretic quantities using a more refined analysis?

**Ethical Concerns:**

["NO or VERY MINOR ethics concerns only"]

**Final Justification:**

My initial opinion on this submission was a bit mixed. I was happy to see that the rebuttal phase satisfactorily addressed my comments (including a technical issue that was actually pretty simple). After reading the other reviews, I decided to raise my score because I think the problem is rather fundamental and the solution is elegant.

There are some remaining unresolved questions that are correctly reported in the submission. I do not think they should prevent this work from being published.

I do not see the lack of experiments as a problem for this type of contribution.

**Limitations:**

Yes

**Quality:**

3

**Strengths And Weaknesses:**

Strengths: An elegant information-theoretic analysis of a simple yet fundamental setting. The upper and lower bounds capture the range of possible dependencies of the regret on the noise variance. The analysis contains some novel ideas and is technically solid and competent. In particular: the loss estimator for the uniform distribution is surprising, and the lower bound cleverly combines classical proofs with information-theoretic tools like the strong data-processing inequality. The overall presentation is clear.

Weaknesses: The upper bound analysis is a rather straightforward application of the exponential weighting technique. The lower bound for symmetric log-concave noise distributions is not uniformly tight.

I think this is an interesting contribution that deserves publication, but perhaps a research note could be a venue more appropriate than NeurIPS.

---

> ### Author Rebuttal · Authors · 2025-07-30
>
> We thank the reviewer for their thoughtful evaluation and constructive feedback. We address their questions below:
>
> - What additional properties of symmetric log-concave noise distributions could capture the right scaling of the regret?
>
> This is an excellent question that touches on the heart of our contribution. We conjecture that the tail behavior determines which regime applies. Indeed, our result shows that for any noise with polynomially decaying tails, $\sigma^2$ scaling is achievable, and already for the Gaussian distribution, which has subexponentially decaying tails (as does any log-concave distribution), better scaling cannot be achieved. Nevertheless, the uniform distribution, which is bounded, achieves the $\sigma$ scaling; better scaling is not achievable for any symmetric log-concave noise distribution.
>
> - Step (c) in display (40) seems to use independence between $\widehat{\ell}\_{t,j}$ and $\mathcal{E}$, which appears to be invalid. Can you comment?
>
> Step (c) in equation (40) applies the Cauchy-Schwarz inequality: $\mathbb{E}[|\widehat{\ell}\_{t,j}|1(\mathcal{E})] \le \sqrt{\mathbb{E}[\widehat{\ell}\_{t,j}^2]}\sqrt{\mathbb{E}[1(\mathcal{E})]}$ for which an independence assumption is not required. The subsequent step (d) then uses the unbiasedness of the estimator and our MSE bounds to complete the analysis. We will clarify this in the revised manuscript.
>
> - The analysis heavily relies on componentwise independence of the noise. Would the analysis of joint noise distributions require a significantly different approach?
>
> Yes, analyzing joint (non-componentwise independent) noise would require significantly different techniques. Our approach fundamentally relies on independence in two key places: the information-theoretic lower bound decomposes the mutual information across experts (equation 58), and the single-letterization argument bounds cumulative information linearly in time (equation 57). With correlated noise, both decompositions fail—we would need multivariate information-theoretic tools and potentially joint estimation across all experts rather than component-wise analysis. This represents an interesting but technically challenging extension of our framework.
>
> - While the upper bound depends on the variance of the MSE, the lower bound depends on an information-theoretic quantity. Can you comment on the feasibiliy of deriving upper bounds depending on similar information-theoretic quantities using a more refined analysis?
>
> Indeed,  achieving upper bounds in terms of $\eta$ would provide a much more unified characterization. The key would be establishing that there exists an unbiased estimator with the MSE bounded by some function of $\eta$. Potentially, this direction would require us to leverage results on relationships between information and estimation quantities such as extensions of the I-MMSE relationship beyond Gaussian channels to general channels. We view this as a compelling direction that could not only close the gap for general noise distributions, but potentially provide a unified characterization that eliminates the need for ad-hoc estimator construction for each noise distribution individually—our upper bounds already exhibit similar scaling with $\eta$ as our lower bounds, suggesting such a unification may be achievable. More precisely, we hope to show that the minimum-variance unbiased estimator achieves MSE that can be upper bounded in terms of information-theoretic quantities like mutual information (in contrast to lower bounds in terms of information measures which are relatively better known). In establishing this relationship, we anticipate uncovering a principled path to optimal estimator construction that would naturally yield the naive estimator $\widehat{\ell}\_t = c\_t$ for Gaussian and Laplace noise (where simple estimation is optimal) and the sophisticated truncated construction of Equation (12) for uniform noise (where careful clipping is required). This would provide both fundamental performance guarantees and constructive guidance for deriving optimal estimators across different noise families.

---

> > ### Comment · Reviewer_vFEj · 2025-08-01
> >
> > Thanks for your competent answers to my questions. I am now siding with Tk3H in thinking that this is a cool result and could be a nice contribution to NeurIPS. So I signal my intention to raise my score from "borderline accept" to "accept" and wait for the replies from the other reviewers.

---

> > > ### Author Response · Authors · 2025-08-06
> > >
> > > Thank you for the positive feedback and for taking the time to engage with our responses! Please let us know if any further clarifications would be helpful.

---

### Official Review · Reviewer_YTLw · 2025-06-30

**Clarity:** 4
**Significance:** 2
**Originality:** 3
**Rating:** 4
**Confidence:** 4

**Summary:**

The authors consider the problem of prediction with expert advice with adversarial losses, where the learner observes noisy versions of the losses which are obtained by perturbing the basic losses with some fixed distribution. The authors consider the specific examples of Gaussian and uniform noise, and establish optimal regret bounds in terms of the standard deviation of the noise in both cases. For more general noise distributions, the authors establish regret upper and lower bounds which are not tight in general, but are valid for all symmetric distributions with sufficiently light tails.

**Questions:**

I would highly appreciate it if the authors could address the points which I raised under "Weaknesses". In particular, regarding the extensions of the results to the bandit setting. I would also appreciate a comment regarding the tightness of the bounds for general symmetric distributions, and the overall technical challenges in the analysis.

**Ethical Concerns:**

["NO or VERY MINOR ethics concerns only"]

**Final Justification:**

Having read the other reviews and comments, I am still leaning towards acceptance, though some valid concerns were raised regarding the similarity to [BK24]. It seems to me like this work does provide interesting extensions to this work which is why I don't view this as a major issue. I therefore maintain my overall score.

**Limitations:**

yes

**Quality:**

3

**Strengths And Weaknesses:**

Strengths:

* The problem studied in this paper is interesting and well-motivated.
* The problem formulation is very simple and easy to understand.
* The paper is written clearly, and the analysis is easy to follow.
* Theorem 1 is a nice way to generalize the regret upper bounds obtained in the paper for all noise distributions.
* Theorem 2 showing a lower bound which depends on the data-processing coefficient of the underlying noise distribution is a nice way to obtain specific lower bounds for certain noise distributions.
* The fact that for some distributions estimating the loss directly via the noisy loss is not the optimal choice (as with the uniform distribution) is surprising and nontrivial.

Weaknesses:

* The fact that the bounds established in the paper are only tight for two given distributions (Gaussian and uniform) makes the overall contribution, in my opinion, a bit lacking. The result would be much stronger if certain distribution-dependent parameters were introduced which would result in tight bounds for the more general family of distributions.

* In terms of technical challenges, the analysis presented in the paper seems quite straightforward, and in particular the general upper bound in Theorem 1 follows from standard arguments in the online learning literature. I think that the most interesting and technically novel part of the analysis is the fact that it is not always optimal to estimate the losses as the noisy losses directly (as with the uniform noise case), which could result in a better dependence on the standard deviation. I think the authors should put more emphasis on this idea compared to other more standard arguments.

* I think the one most natural questions given the results presented in this paper concerns bandit feedback, which is not mentioned in the discussion section of the paper. It would seem to me that the technical difficulty in extending those results to the bandit setting arise when estimating the losses via importance sampling, which would possibly result in highly negative loss values on top of the noise being unbounded. Have the authors consider this extension? I think this paper would highly benefit from such a generalization as the results just for the full-information setting seem to me as more of an initial step in the right direction.

* The estimators provided for the algorithm require knowledge of the noise standard deviation $\sigma$. It would seem possible, though perhaps more complicated, to achieve similar regret bounds with adaptive algorithms which do not have such prior knowledge.

* The lower order terms in the bounds written in Eqns (11), (14) are a bit confusing - rather than $o(n)$ which only signifies sublinear dependence on $n$, the bounds actually decay to zero as a function of $n$, so it should be $o(1)$.

---

> ### Author Rebuttal · Authors · 2025-07-30
>
> We thank the reviewer for their detailed evaluation and insightful comments.
>
> - On Tightness of the Bounds: We are very interested in obtaining a more precise characterization of the regret. As we mentioned in the reply to question 4 of reviewer vFEj, this would likely require a deeper understanding of the relationships between the estimation-theoretic and information-theoretic quantities than already established - such relationships are an active area of research. In our paper, we derive novel upper and lower bounds on the regret that apply to any log-concave symmetric noise. We further demonstrate that neither bound is vacuous - the Gaussian distribution satisfies the upper bound with equality, while the uniform distribution attains the lower bound. As highlighted by both reviewers Tk3H and YTLw, our uniform noise estimator is novel and represents a key insight—it demonstrates how the distributional knowledge can be leveraged to improve upon naive estimation in real-time decision-making. In our paper, we also demonstrated an example (lines 286-290) of an additive noise distribution which despite have $\sigma^2$ variance, leads to $\eta = 1$ and the noisy regret being exactly equal to the noiseless regret---this shows that simply restricting variance does not lead to a universal result. Upon imposing some more structure on noise density (log-concavity) we are able to achieve nearly-optimal characterization of regret; the question of both tightening the bounds and achieving such a characterization for an even larger class of noise distributions over symmetric log-concave remains an exciting open direction.
>
>
> - On Technical Challenges: In the revised manuscript, we will highlight the novelty beyond the uniform noise case. Note also that the lower bound (tight for the uniform noise) holds for any symmetric log-concave noise. Our approach to the converse encompassing such a wide family of distributions is novel - we leverage the strong data processing inequality, and we apply the log-concavity of the noise to bound the strong data processing constant in terms of the variance universally within that class.
>
>
> - On Bandit Extensions: This is an excellent and challenging question that we have considered. The bandit setting would indeed compound two sources of difficulty: importance sampling can create estimates with high variance, and additive noise makes this even worse since we might observe very negative values. However, we believe this is surmountable using techniques to control for these challenges. For instance, one could use truncated importance sampling (clipping estimates to reasonable ranges) combined with bias correction, or ideas for variance reduction. The key “separation principle” insight as in Theorem 1 would likely extend, but the specific estimator constructions would need to account for the importance sampling structure.
>
>
> - On Adaptive Algorithms: You raise an important practical limitation. The fundamental challenge is that our optimal estimators (particularly for uniform noise) require precise distributional knowledge. However, we believe there are promising approaches to achieve similar rates adaptively. One direction is using online parameter estimation—for instance, simultaneously learning the noise variance while running the expert algorithm, potentially using tricks to handle the exploration-exploitation tradeoff. Another approach might be developing estimators that are robust across parameter ranges, though this could sacrifice some optimality. The theoretical question of whether adaptive algorithms can match the optimal rates remains an intriguing open problem.
>
>
> - On lower-order terms: Thank you for pointing this out, we will correct it in the updated manuscript.

---

> > ### Comment · Reviewer_YTLw · 2025-08-04
> >
> > Thank you for the detailed response.
> >
> > While my concerns have been mostly addressed, I am currently inclined to maintain a borderline score. A partial reason is the concern regarding the originality of this work compared to [BK24], as raised by reviewer rqtx. While I partially agree that the two works are similar, I recognize that there are several novelties in this work which are of value, and therefore I do not think those similarities warrant a rejecting score. I will finalize my decision after discussing with the other reviewers.

---

> > > ### Author Response · Authors · 2025-08-06
> > >
> > > We appreciate your thoughtful consideration and understanding of the technical novelties in our work beyond [BK24]. We remain committed to addressing any presentation concerns if the paper is accepted, as outlined in our responses. Thank you for your willingness to engage in discussion; we're happy to provide any additional clarifications that would be helpful.

---

### Official Review · Reviewer_rqtx · 2025-07-02

**Clarity:** 3
**Significance:** 3
**Originality:** 3
**Rating:** 5
**Confidence:** 3

**Summary:**

The paper considers prediction with expert advice where the losses are observed with stochastic noise. Upper bounds are shown for bounded losses using an EXP3-like algorithm (exponential weights + unbiased estimates) in three noise settings: Gaussian, uniform and log-concave. Lower bounds are shown for binary losses in terms of the information processing constant of the noising mechanism.

**Questions:**

I'd like to hear the authors view on submitting a paper with pages of text copied compared with their own earlier publication.

**Ethical Concerns:**

["NO or VERY MINOR ethics concerns only"]

**Final Justification:**

This paper has multiple pages of nearly identical text with a previous paper. It also has new results that significantly go beyond that earlier paper. The authors propose to remove the overlap. As I wrote to them below, I think it is in fact simpler to acknowledge that this is an extended version of their earlier paper, thus making the overlap non-problematic. This is not difficult or time consuming to implement, so it can be expected of the authors.

With that problem out of the way. We then end up with a collection of interesting new results, about which I am positive. I have updated the score accordingly.

**Limitations:**

yes

**Paper Formatting Concerns:**

-

**Quality:**

2

**Strengths And Weaknesses:**

Clarity:
The paper is clear.

Quality:
The problem is interesting. The results are an elegant characterization of the minimax regret.

Significance:
The problem of noisy feedback is reminiscent of (but simpler than) the information bottleneck present in bandits. Compared to [BK24], the new parts are the extension to unbounded noise, and the adaptation of Theorem 1. These are relatively moderate contributions.


Originality:
The work is an extended version of the ISIT publication [BK24]. Not a follow-up as is claimed. I say this because the text has significant overlap. If these are not the same authors, then this is clear plagiarism. Yet even if they are the same authors, this relation (of being the extended version of) is not mentioned in the paper. It is certainly pushing the boundary of the Neurips submission guideline "Submissions that are identical (or substantially similar) to versions that have been previously published".

---

> ### Author Rebuttal · Authors · 2025-07-30
>
> We thank the reviewer for their careful evaluation and appreciate the opportunity to address their concerns about originality and our relationship to prior work.
>
> We want to clarify our substantial technical extensions beyond the work in [BK24]. As explicitly stated in our paper (lines 171-178): "[BK24] recently studied prediction with noisy expert advice, but their analysis had several limitations that the current work addresses. First, this paper provides a comprehensive treatment for both bounded and unbounded losses—this is a critical consideration when dealing with additive noise with potentially heavy tails. Second, while they presented (without proofs) an ad-hoc lower bound for Gaussian noise, this work derives it from a unified framework of Theorem 2. Third and most importantly, this work establishes tight characterizations for a wide class of additive noise channels, including the nearly-tight bounds for uniform and general symmetric log-concave distributions that do not follow from their work."
>
> We now elaborate on this. Our technical contributions represent advances in several key areas:
>
>
> - New ideas in Optimal Estimation: Our discovery that optimal estimation can fundamentally differ from naive approaches represents a significant theoretical insight. As other reviewers have noted, "the loss estimator for the uniform distribution is surprising" and demonstrates that naive estimation $\widehat{\ell}_t = c_t$ can be suboptimal by factors up to $\sqrt{\sigma}$. This reveals structure about when and how distributional knowledge affects learning performance.The decomposition of noisy regret into estimation error plus optimal control (Theorem 1) provides a general principle with applications beyond this specific setting, offering new perspectives on the connection between estimation theory and online learning.
>
>
> - Comprehensive Analysis of Log-Concave Distributions: Our nearly-tight characterization of the broad class of symmetric log-concave noise distributions, with matching upper and lower bounds for specific cases (particularly the Laplace and uniform distributions), represents a theoretical contribution that significantly advances understanding of this problem.
>
> - Extension to Unbounded Noise: The treatment of unbounded distributions like Laplace noise required new technical approaches including an analysis of "bad events" and probability bounds that were not addressed in preliminary work.
>
>
> - On Textual Overlap Concerns: We acknowledge the reviewer's concern about textual similarity in certain sections. Upon reflection, we recognize that some passages in our introduction and problem setup as well as related work overlap with [BK24]. In particular, there is overlap in the Introduction section which elaborates on the problem (Lines 13-15, 21-26 that explain the vanilla experts problem, and lines 42,43,49-52, 56-63 that introduce the technical details of prediction with noisy experts problem as well as precise definitions) and overlap in the Section 3 (lines 142-170, which is an exposition of related work). This occurred because both works necessarily establish the same fundamental problem formulation and mathematical framework. This also reflects standard practice when extending preliminary results - foundational mathematical definitions and problem formulations necessarily share common elements. Similarities in the Sections on technical content include Lemma 1 (lines 809-810), Definition 1 (lines 251-252), and the ensemble in (48)---these are all standard results/definitions taken from the literature and therefore are similar.
>
>
> We finally want to address some important points:
>
> -   Substantial New Content: The overwhelming majority of our paper consists of entirely new technical material. Our theoretical analysis, novel estimator constructions, information-theoretic converse for log-concave distributions, and applications to new noise models represent contributions that extend far beyond any preliminary work.
>
>
> -  Path Forward: We sincerely apologize for our oversight in writing. If our work is accepted, we will revise the intro and related work sections to minimize any textual similarities while preserving the necessary technical foundations. In particular, we will rewrite the introduction by motivating the importance of understanding the limits of real-time decision-making where the losses are corrupted by an additive noise whose distribution does not follow previously considered examples of binary channel models (BSC, BEC, AWGN) to encompass continuous distributions that arise naturally in practical signal processing and quantization applications. As a prominent example, uniform additive noise emerges in quantization-limited systems where continuous signals must be discretized with finite precision—ubiquitous in analog-to-digital conversion, communication-constrained IoT sensor networks, and federated learning with limited-precision arithmetic. Our discovery that uniform noise requires fundamentally different optimal estimation techniques represents a surprising theoretical insight with immediate practical implications for these quantization-dominated environments in order to achieve optimal regret. More broadly, symmetric log-concave noise distributions provide a rich framework encompassing Laplace distributions, truncated Gaussian models and other distributions that better capture measurement errors compared to pure Gaussian assumptions, making our theoretical framework useful for understanding fundamental performance limits across these diverse practical scenarios.
>
>
> We appreciate the reviewer's diligence in evaluating our work and welcome any further clarification they may need about our technical contributions or relationship to prior work.  We also welcome any advice on how to best address their concerns.

---

### Official Review · Reviewer_tNvt · 2025-07-03

**Clarity:** 2
**Significance:** 3
**Originality:** 2
**Rating:** 4
**Confidence:** 3

**Summary:**

This paper addresses the problem of prediction with expert advice under additive noise, and aims to achieve performance comparable to the best fixed expert in hindsight on the uncorrupted loss. The main contribution is a characterization of performance limits via sharp regret bounds for various noise models. Specifically, the authors show that for Gaussian noise with variance $\sigma^2$, regret scales as $\Theta(\sqrt{(1 + \sigma^2)n \log m})$, while for uniform noise in $[−\sigma, \sigma]$, it scales as $\Theta(\sqrt{(1 + \sigma)n \log m})$. For the broader class of symmetric log-concave noise distributions, bounds are provided to be between these regions, i.e.,  $\Omega(\sqrt{(1 + \sigma)n \log m}) \leq Reg \leq O(\sqrt{(1 + \sigma^2)n \log m})$. These theoretical guarantees are derived by combining an enhanced exponential weights algorithm that leverages an unbiased estimator and information-theoretic lower bounds. This work overcomes prior limitations by providing a unified framework for both bounded and unbounded losses and establishestight characterizations for a wide range of additive noise channels.

**Questions:**

1) What is the reason for the persistent gap in regret bounds for general symmetric log-concave noise? Are there any specific technical barriers preventing tighter characterization? Does tighter bounds require a different analytical approach?

2) What are the main conceptual and/or technical challenges in developing high-probability guarantees for prediction with expert advice under additive noise, beyond the expected regret analysis?

3) What are the fundamental obstacles to extending your framework to alternative loss functions (beyond linear losses) and to infinite expert classes?

4) Though no experiments are included, which theoretical aspects do you believe would be most insightful to validate empirically and the practical implications such validation might yield?

**Ethical Concerns:**

["NO or VERY MINOR ethics concerns only"]

**Final Justification:**

I am keeping my score, which remains a positive.

The authors clarified that while the bounds are not universally tight for all symmetric log-concave distributions, they are "nearly-tight" because the upper bound is achieved by Gaussian noise and the lower bound by uniform noise, both of which are log-concave. This highlights the paper's key insight: the shape of the noise distribution, not just its variance, fundamentally impacts regret scaling. The exponential weights algorithm's effectiveness with unbiased estimators, quantifying performance degradation through estimation error (MSE), is now clearer.

The paper remains purely theoretical. While the authors stated theoretical plots suffice, empirical experiments would strengthen practical applicability. The analysis is confined to expected regret, linear losses, and finite expert classes, though extensions are noted as future work.

I assign high weight to the enhanced theoretical clarity regarding noise-dependent regret scaling and the strong theoretical framework. The lack of empirical validation and current scope are given moderate weight, as the theoretical insights are valuable on their own.

**Limitations:**

yes

**Paper Formatting Concerns:**

no major issues

**Quality:**

2

**Strengths And Weaknesses:**

# Strengths

1) The paper tackles the practical limitation of perfect feedback in online learning, focusing on additive noise in real-world scenarios like sensor inaccuracies and autonomous driving.

2) It provides a comprehensive characterization of fundamental performance limits for prediction with expert advice under various additive noise models.

3) It clearly demonstrates how different noise characteristics affect regret scales, noting superlinear regret growth with increasing noise intensity and a smooth transition to the noiseless case as noise diminishes.

4) Guarantees are derived using an enhanced exponential weights algorithm with an unbiased estimator (linking regret to estimation MSE) and information-theoretic lower bounds based on the strong data-processing constant, which quantifies information preservation.

# Weaknesses

1) While "nearly-tight," the bounds for the broad class of symmetric log-concave noise are not perfectly tight.

2) The paper does not include experiments or empirical results, meaning the theoretical findings are not validated computationally.

3) The paper itself identifies areas for future work which highlight current limitations, such as focusing only on expected regret analysis (lacking high-probability guarantees for risk-sensitive applications) and limiting the framework to linear losses and finite expert classes.

---

> ### Author Rebuttal · Authors · 2025-07-30
>
> We thank the reviewer for their detailed evaluation and insightful questions. We address each point below:
>
>
> 1. Gap in Log-Concave Bounds. We are very interested in obtaining a more precise characterization of the regret. This would likely require a deeper understanding of the relationships between the estimation-theoretic and information-theoretic quantities than already established - such relationships are an active area of research. In our paper, we derive novel upper and lower bounds on the regret that apply to any log-concave symmetric noise. We further demonstrate that neither bound is vacuous - the Gaussian distribution satisfies the upper bound with equality, while the uniform distribution attains the lower bound. We conjecture that the tail behavior determines which regime applies. Indeed, our result shows that for any noise with polynomially decaying tails, $\sigma^2$ scaling is achievable, and already for the Gaussian distribution, which has subexponentially decaying tails (as does any log-concave distribution), a better scaling cannot be achieved. Nevertheless, the uniform distribution, which is bounded, achieves the $\sigma$ scaling; a better scaling is not achievable for any symmetric log-concave noise distribution.
>
>
> 2. High-Probability Guarantees. We are optimistic that high-probability bounds would be quite achievable using standard concentration inequality techniques. The main technical steps would involve:
> - Applying martingale concentration inequalities (like Azuma-Hoeffding) to the noise-corrupted loss estimates
> - Using union bounds over the finite expert set and time horizon
> - Carefully tracking how estimation errors concentrate around their means
> - Our separation principle already decomposes the problem into estimation + exponential weights components, and both parts have well-developed high-probability analysis tools. This represents a tractable extension of our current framework.
>
>
> 3. Extensions to Non-Linear Losses and Infinite Experts. For infinite expert classes, this extension seems quite feasible using techniques from statistical learning theory. The approach would involve replacing our finite $\log m$ term with covering number bounds or Rademacher complexity, and further using metric entropy arguments to bound the effective number of experts. The estimation component of regret would be unchanged, while the exponential weights component of regret could be handled by the aforementioned techniques. For non-linear losses, while more challenging, we see promising directions: The separation principle could possibly be extended using appropriate bias-variance decompositions and techniques from online convex optimization might bridge the gap between estimation and regret.
>
>
> 4. Empirical Validation Priorities. Thank you for raising this interesting question! We believe the most insightful empirical validation would focus on several key areas. Theoretical prediction verification would involve testing our scaling laws—in particular plotting regret vs. noise parameter $\sigma$ to confirm the scaling with regret for Gaussian, Uniform and Laplace distributions. Estimator comparison studies would be particularly compelling, especially demonstrating that our uniform noise estimator (Equation 12) significantly outperforms naive estimation $\widehat{\ell}\_t = c\_t$, which should show dramatic improvements and validate our theoretical insight about optimal estimation. Robustness analysis to test model misspecification scenarios (e.g., assuming Gaussian when noise is heavy-tailed) and performance under parameter uncertainty would be insightful. Finally, a preliminary real-world motivated study could include asset price prediction in financial markets, assuming noisy observations.
>
>
> We appreciate the reviewer's constructive suggestions and believe the extensions suggested by them represent exciting and achievable directions for future work.

---

> > ### Comment · Reviewer_tNvt · 2025-08-05
> >
> > I appreciate the authors' rebuttal and have some follow up comments.
> >
> > - The paper states that the characterization for symmetric log-concave noise (Eq.6) is "not perfectly tight" and that the lower bound (Eq. 29) is "not tight in general". However, the rebuttal claims that the derived upper and lower bounds for Gaussian and uniform noise are "sharp" and represent the "optimal scaling for additive noise channels in general". Could you clarify this seeming contradiction? Specifically, if the broader class of symmetric log-concave distributions leads to a gap between the upper and lower bounds, how can the bounds be considered "optimal scaling for additive noise channels in general," especially when the argument is made that "a better scaling is not achievable for any symmetric log-concave noise distribution" due to the very nature of noise shape?
> >
> > - You express optimism that high-probability bounds are achievable using standard concentration inequality techniques. Given that the paper focuses on adversarial loss sequences, what unique challenges or complexities arise when extending the exponential weights algorithm with unbiased estimators to provide high-probability guarantees (e.g., using martingale concentration inequalities like Azuma-Hoeffding) in this noisy and adversarial setting, compared to standard noiseless online learning problems?
> >
> > - You state that the extension to non-linear losses and infinite expert classes seems "quite feasible" and that the "component of regret due to noise would be unchanged". Could you briefly explain why the noise-induced component of the regret would remain invariant, even if the loss function is no longer linear or the expert space becomes infinite? What fundamental property of the additive noise model or the regret definition ensures this robustness?

---

> > > ### Author Response · Authors · 2025-08-05
> > >
> > > We thank the reviewer for engaging with our rebuttal and the follow up questions, which we address below.
> > >
> > > - We believe there may be a misunderstanding here. We cannot locate the phrase "optimal scaling for additive noise channels in general" in either our paper or rebuttal. Our actual statements were more nuanced: we showed that Gaussian distributions achieve $\sigma$ scaling and that "a better scaling cannot be achieved" for Gaussian noise specifically, while uniform distributions achieve $\sqrt{\sigma}$ scaling and a better scaling is not achievable for any symmetric log-concave noise distribution (including the uniform noise). We moreover demonstrate that for any noise distribution in the log-concave family of distributions, the optimal scaling of regret lies between $\sqrt{\sigma}$  and $\sigma$.
> > >
> > > We did not claim universal optimality across all additive noise channels. Rather, our contribution is demonstrating that different noise shapes within the log-concave class can achieve fundamentally different scaling behaviors, with our bounds being tight for the specific cases we analyzed. If we've inadvertently created this impression through unclear wording, we'd be grateful for the reviewer's guidance on which specific passages led to this interpretation so we can clarify our intended meaning.
> > >
> > >
> > >
> > > -  The extension to high-probability guarantees involves accounting for two distinct sources of randomness: the randomness of the algorithmic decisions themselves, and the randomness induced by the noisy channel. The first source, obtaining high-probability regret bounds for the exponential weights algorithm, is well-established in the literature. The second source, which represents the main departure from the classical noiseless setting, can likely be handled using standard concentration inequalities for additive noise, leveraging the favorable tail behavior of log-concave distributions. The two sources could potentially be combined via union bounds, though careful analysis of the conditioning between algorithmic and noise randomness would be required, since the algorithm's random choices at time t depend on the noisy observations from previous rounds. While this interaction introduces technical complexity beyond standard noiseless analysis, the overall approach appears tractable within our separation principle framework.
> > >
> > >
> > > -  We should clarify this statement. The noise-induced regret component remains invariant in the sense that even for infinite experts, the estimation error $\mathbb{E}||\widehat{\ell}_t-\ell_t||^2$ depends only on the noise distribution and loss boundedness, not on the number of experts. The complexity of handling infinite experts appears in the "algorithmic" component (replacing $\log m$ with covering numbers), while the noise analysis remains unchanged.
> > > We thank the reviewer for pushing us to be more precise about these technical distinctions.

---

### Note · Authors · 2025-08-13

We deeply appreciate the constructive and thoughtful engagement from all reviewers. The discussion has highlighted several key points we wish to emphasize for the AC's consideration.

Our main and novel contributions include (1) comprehensive treatment of unbounded noise distributions, (2) novel information-theoretic framework for achieving nearly-tight characterizations for symmetric log-concave distributions, (3) the separation principle decomposing regret into estimation + control components, and (4) discovery of fundamental differences in optimal estimation strategies across noise types (notably for uniform noise), with immediate implications for quantization-limited systems.

Technically, we clarified that Gaussian and uniform noise attain our upper and lower bounds respectively; for general symmetric log-concave noise, optimal scaling lies between these regimes, dictated by tail behavior.

Several extensions to the current framework were proposed by the reviewers, and we believe these open up very exciting avenues for future work. High-probability bounds appear tractable using martingale concentration inequalities and our separation principle, though the interaction of algorithmic and noise randomness requires care. Extensions to infinite experts and non-linear losses preserve the noise-induced regret term, with complexity changes confined to the algorithmic component. Bandit extensions, while harder due to compounding noise and importance-sampling variance, seem approachable via clipping and variance reduction.

We acknowledge that some bounds are “nearly tight” and see unifying them—potentially through information-theoretic upper bounds—as an exciting open direction. Several reviewers supported acceptance, highlighting novelty in the uniform noise estimator and breadth of scope. We believe the paper offers both fundamental insight and practical relevance, and are committed to strengthening clarity in the final version.
We acknowledge feedback regarding presentation concerns in certain introductory and foundational paragraphs. If accepted we will rewrite these paragraphs to enhance clarity and emphasize the novelty of our work while preserving necessary technical foundations.

---

### Decision · Program_Chairs · 2025-09-17

**Decision:**

Accept (poster)

**Comment:**

This paper addresses the problem of prediction with expert advice under additive noise, and aims to achieve performance comparable to the best fixed expert in hindsight on the uncorrupted loss. The main contribution is a characterization of performance limits via sharp regret bounds for various noise models.

All reviewers are positive about the contributions of the paper with a consensual rating range (4-5).
An issue raised by a reviewer is an important textual overlap in text and setting presentation with a previous reference.
The contributions themselves are new and of independent interest as noted by all reviewers.
The authors promised to address this in the final version: "If accepted, we will revise the introduction and related work sections to minimize textual similarities while preserving necessary technical foundations."